# EFFICIENT INTERACTIVE PREFERENCE LEARNING IN EVOLUTIONARY ALGORITHMS: ACTIVE DUELING BANDITS AND ACTIVE LEARNING INTEGRATION

## ABSTRACT

Optimization problems find widespread use in both single-objective and multi-objective scenarios. In practical applications, users aspire for solutions that converge to the region of interest (ROI) along the Pareto front (PF). While the conventional approach involves approximating a fitness function or an objective function to reflect user preferences, this paper explores an alternative avenue. Specifically, we aim to discover a method that sidesteps the need for calculating the fitness function, relying solely on human feedback. Our proposed approach entails conducting a **direct preference learning** facilitated by an active dueling bandit algorithm. The experimental phase is structured into three sessions. Firstly, we accsess the performance of our active dueling bandit algorithm. Secondly, we implement our proposed method within the context of **Multi-objective Evolutionary Algorithms (MOEAs)**. Finally, we deploy our method in a practical problem, specifically in protein structure prediction (PSP). This research presents a novel interactive preference-based MOEA framework that not only addresses the limitations of traditional techniques but also unveils new possibilities for optimization problems.

## 1 INTRODUCTION

In optimization problems, algorithms typically converge to the Pareto front (PF), yet users aim for convergence in their specific region of interest (ROI). To bridge this gap, constructing a fitness function to capture user preferences is common, involving considerations of multiple metrics, especially in multi-objective optimization problems (MOPs) (Miettinen & Mäkelä, 2000; Li et al., 2019; Deb & Kumar, 2007; Branke et al., 2015; Tomczyk & Kadziński, 2019; Deb et al., 2010; Chugh et al., 2015; Chen et al., 2021; Kadziński et al., 2020). However, a significant challenge arises when the ROI lacks a fitness function or the fitness function is hard to express. The fitness-based approach struggles without a baseline to learn and evaluate preference accuracy. To address this, given the inevitability of human input in the ROI, our approach explores the possibility of focusing on global optima through user preferences.

This paper centers on direct **preference-based evolutionary multi-objective optimization (PBEMO)**, where reliance on human feedback is crucial for cost-effective and accurate exploration in the absence of a fitness function. Existing solutions, like the dueling bandit approach (Yan et al., 2022; Bengs et al., 2021; Sui et al., 2018) and reinforcement learning (RL) (Myers et al., 2023; Rafailov et al., 2023), offer insights but fall short in addressing the challenges of expensive sampling and consultation. Striking a balance between accuracy and consultation frequency is vital, given that excessive queries may yield inaccurate feedback. The dueling bandit method, leveraging pairwise comparisons for optimal arm identification, emerges as a simple and effective approach. This paper explores a dueling bandit algorithm adept at managing sampling costs to tackle challenges in PBEMO preference learning .

The current state of PBEMO faces three main challenges. **Firstly**, traditional preference learning (PL) in PBEMO algorithms relies on an indirect approach, using a fitness function to capture user preferences, which proves less effective in fitness-free scenarios (Section 3.3). **Secondly**, the practicality of sampling and consultation introduces a substantial expense. While studies acknowledge

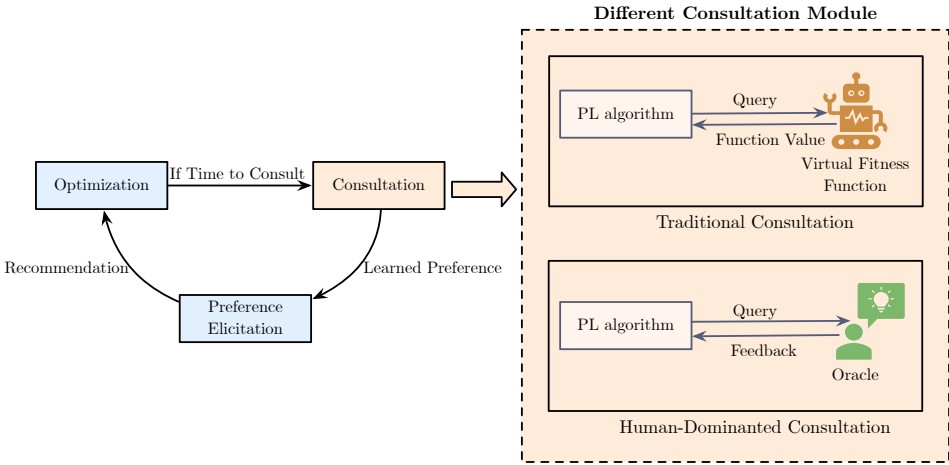

Figure 1: A deception of difference between traditional PBEMO and our proposed human-dominated PBEMO architecture

that repetitive queries may yield inaccurate feedback (Hejna & Sadigh, 2023), quantifying optimal query times remains unaddressed. **Lastly**, existing fitness-based PBEMO algorithms lack a strict mathematical regret bound.

To overcome these challenges, we introduced ***RUCB-AL***, an active preference learning algorithm based on dueling bandits acting as a decision maker (DM) in PBEMO structure. Our baseline algorithm is the well-known RUCB (Zoghi et al., 2014a). The integration of RUCB with active learning (Settles, 2009; Ren et al., 2021) aims to control the budget of query times while ensuring accurate preference prediction. Simultaneously, we proposed an efficient mechanism to decide when to start or stop consultation and optimally select incumbent solutions for DM. The architecture of direct PBEMO consists of three main components, Fig. 1: **Optimization Module**, employing MOEAs like dominance-based EA (e.g., NSGA-II (Deb et al., 2002a)), decomposition-based EA (e.g., MOEA/D (Zhang & Li, 2007)), and indicator-based EA (e.g., R2-IBEA (Zitzler et al., 2004)); **Consultation Module**, tailored for *"active pairwise comparison"* by balancing random search and greedy search; **Preference Elicitation Module**, which reprocesses the **virtual fitness** (Section 2.4) function by accumulating past recommendations.

In the empirical study, we begin by validating the active learning capability of our proposed method through a comparative analysis with other pairwise preferential module. Subsequently, we apply our proposed method on MOP test suites (i.e., ZDT (Deb et al., 2002b), DTLZ (Zitzler et al., 2000), WFG (Huband et al., 2006)), and assess its performance against peer algorithms. Finally, we extend our algorithm to address a real-world problem, specifically protein structure prediction (PSP) (Zhang et al., 2023).

In summary, we have these three main contributions:

- We introduced a direct PBEMO framework that directly learns the global optima from human feedback, applicable not only to single-objective problems (SOPs) but also to MOPs by integerating it with three categorical MOEAs.

- We incorporated active learning in dueling bandit, enabling the quantification of the budget for sampling and consultation. Our active dueling bandit has a regret bound of $\mathcal{O}(K)$.

- Beyond validation on basic benchmark problems, we demonstrate the practical applicability of our proposed method by implementing it on a practical problem, PSP. The application showcases the versatility of effectiveness of our approach in addressing practical problems.

The related work is available in Appendix A.1.

## 2 PROPOSED METHOD

### 2.1 PROBLEM STATEMENT

The MOP (Deb, 2001) considered in this paper is defined as follows:

$$\min_{subject\ to\ \mathrm{x}\in\Omega} \mathbf{F}(\mathrm{x}) = (f_1(\mathrm{x}), f_2(\mathrm{x}), \ldots, f_m(\mathrm{x}))^\top. \tag{1}$$

where a solution x represents a vector of $n$ dimension variables: $\mathrm{x} = (x_1,\ x_2, \ldots,\ x_n)^T$ and $\mathbf{F}(\mathrm{x})$ denotes an $m$-dimensional objective vector where each objective function can be either minimized or maximized. For the purpose of this paper, we focus on the minimization problem.

The feasible region $\Omega$ resides within the decision space $\mathbb{R}^n$, while the mapping collection $\mathbf{F} : \Omega \to \mathbb{R}^m$ corresponds to the objective space $\mathbb{R}^m$. When considering two randomly chosen solutions, $\mathrm{x}_1$ and $\mathrm{x}_2$, from $\Omega$, we say that $\mathrm{x}_1$ dominates $\mathrm{x}_2$ if $f_i(\mathrm{x}_1) \leq f_i(\mathrm{x}_2)$ holds for all $i \in \{1, 2, \ldots, m\}$. A solution $\mathrm{x} \in \Omega$ is deemed Pareto-optimal if there is no $\mathrm{x}' \in \Omega$ that dominates x. The collection of all Pareto-optimal solutions forms the Pareto-optimal set (PS).

In addition to the decision variable space, the objective functions define a multidimensional space known as the objective space, denoted as $Z$. For each solution x in the decision variable space, a corresponding point $\mathbf{F}(\mathbf{x}) = \mathbf{z} = (z_1, z_2, \ldots, z_m)^T$ exists in the objective space. The objective space associated with the PS is referred to as PF.

### 2.2 OPTIMIZATION MODULE

The evolutionary multi-objective (EMO) algorithm is one of the most commonly used and efficient methods for solving MOPs. EMO algorithms drive the feasible solutions to the PF of test problems. Up-to-date EAs can be classified into three types: domination-based EA (e.g., NSGA-II (Deb et al., 2002a)), decomposition-based EA (e.g., MOEA/D (Zhang & Li, 2007)) and indicator-based EA (e.g., IBEA (Zitzler et al., 2004)). NSGA-II, MOEA/D, and IBEA are our baseline EMO algorithms.

However, providing a set of solutions as close to the Pareto front as possible may not always meet user expectation. In this paper, we focus on PBEMO algorithms which provide the user with a specific solution or a cluster of soltuions close to ROI. With the help of consultation module (Section 2.3), these EAs can reach the interested region.

In each geneartion, we will gather a population of solutions. However, it's not sensibel to feed the consultation module with the whole population. Because solutions share different uncertainty. Not every one is promising to be global optima. We select 10 incumbent solutions from the whole population using a virtual fitness function $V_s$, equation (9). We map the incumbent solution to arms in dueling bandit by calculating the pairwise winning probability $p_{ij} = \sigma(V_s(\mathbf{z}_i) - V_s(\mathbf{z}_j))$ while in the first round $p_{ij} = 1/2$. In this paper, the logistic probability modle $\sigma(x) = 1/(1 + \exp(-x))$ is utilized, which is the common choice in related researches.

### 2.3 CONSULTATION MODULE

Given that humans find it easier to compare two objects and identify which is better (Li et al., 2020b), our proposed method leverages pairwise preference learning in the consultation module. Specifically, it employs direct fitness-free preference learning through the use of dueling bandits.

#### 2.3.1 BASIC DEFINITION

Our proposed method of consultation module is built upon RUCB (Zoghi et al., 2014a). To our knowledge, our method is the first to integrate dueling bandit with active learning. We consider a dueling bandit with $K(K \geq 2)$ arms, denoted by $\mathcal{A} = \{1, 2, \ldots, K\}$. In each round $t > 0$, a pair of arms $(a_i, a_j)$ is chosen for a noisy comparison by users. The comparison result is 1 if $a_i$ is preferred over $a_j$, and the result is 0 vice versa. We assume the user preference is consistent and stationary over time. The distribution of comparison outcomes is characterized by the preference matrix $\mathbf{P} = [p_{ij}]_{K \times K}$, where $p_{ij}$ (Section 2.2) denotes the probability of arm $i$ preferred over arm $j$, $p_{ij} = \mathbb{P}\{a_i \succ a_j\}$, $i, j = 1, 2, \ldots, K$. Also $p_{ij} + p_{ji} = 1$, and $p_{ii} = \frac{1}{2}$. Arm $i$ is said to beat $j$ if

---

**Algorithm 1** RUCB-AL (Relative Upper Confidence Bound in Active Learning)

---

**Input:** $\kappa$, $T \in \{1, 2, \ldots\} \cup \{\infty\}$, oracle $\mathcal{O}$, budget $M$.

1: $W = [w_{ij}] \leftarrow \mathbf{0}_{K \times K}$ //2D array of wins: $w_{ij}$ is the number of times $a_i$ beat $a_j$.
2: $num\_query = 0$.
3: **while** $num\_query \leq M$ **do**
4:     **for** $t = 1, \ldots, T$ **do**
5:         $p_{\min} = \frac{1}{Kt^{\kappa}}$.
6:         Calculate $\hat{\mathbf{P}}$ according to equation (4), where $\hat{p}_{ii} \leftarrow \frac{1}{2}$ for each $i = 1, \ldots, K$.
7:         Calculate $\mathbf{U}$ according to equation (5).
        // All operations are element-wise; $\frac{x}{0} := 0.5$ for any $x$.
8:         $\mathcal{C} \leftarrow \{a_c | \forall j : u_{cj} > \frac{1}{2}\}$.
9:         **if** $\mathcal{C} = \emptyset$ **then**
10:            Pick $c$ from $\{1, 2, \ldots, K\}$ according to the distribution of $p(a_c)$:

$$p(a_c) = p_{\min} + (1 - Kp_{\min}) \frac{\sum_{j=1}^{K} L(\hat{p}_{cj})}{\sum_{i=1}^{K} \sum_{j=1}^{K} L(\hat{p}_{ij})} \tag{3}$$

11:         **else**
12:            Pick $c$ from $\mathcal{C}$ according to equation (3).
13:         **end if**
14:         $d \leftarrow \arg\max_j u_{cj}$, with ties broken randomly.
        Moreover, if there is a tie, $d$ is not allowed to be equal to $c$.
15:         **if** $a_c$ was compared to $a_d$ queried previously **then**
16:            Reuse its previously used query result.
17:         **else**
18:            Query oracle $\mathcal{O}$ for the comparison result arm $a_c$ over $a_d$.
19:            $num\_query \leftarrow num\_query + 1$
20:         **end if**
21:         Increment $w_{cd}$ or $w_{dc}$ depending on which arm wins.
22:     **end for**
23: **end while**

**Output:** An arm $a_c$ that beats the most arms, i.e., $c$ with the highest Copeland score $\zeta_c$.

---

$p_{ij} > \frac{1}{2}$. In this paper, $\hat{\mathbf{P}} = [\hat{p}_{ij}]_{K \times K}$ (equation (4)) denotes the predicted preference matrix, where $\hat{p}_{ij}$ denotes the predicted preference probability.

In the traditional RUCB, the algorithm assumes the existence of Condorcet winner (Urvoy et al., 2013), an arm that has a probability of winning against all other arms greater than $\frac{1}{2}$. However, this requirement may not always be satisfied in practice. So we give the following definition:

**Definition 1.** In this paper, we assume that there exists Copeland winner (Urvoy et al., 2013). An arm $i$ is said to be Copeland winner when:

$$a^* = \arg\max_{i \in \mathcal{A}} \sum_{j \neq i, j \in \mathcal{A}} \mathbb{I}\{p_{ij} > \frac{1}{2}\} \tag{2}$$

where $\mathbb{I}\{p_{ij} > \frac{1}{2}\}$ is the indicator function, $a^*$ denotes the optimal arm among all the $K$ solutions. The Copeland score is defined as $\sum_{j \neq i, j \in \mathcal{A}} \mathbb{I}\{p_{ij} > \frac{1}{2}\}$, and the normalized Copeland score is $\zeta_i = \frac{1}{K-1} \sum_{j \neq i, j \in \mathcal{A}} \mathbb{I}\{p_{ij} > \frac{1}{2}\}$. Let $\zeta^*$ be the highest normalized Copeland score, $\zeta^* = \max_{i \in \mathcal{A}} \zeta_i = \zeta_{a^*}$. The cumulative regret up to round $T$ is defined $R_T = \sum_{t=1}^{T} r_t = \zeta^* T - \frac{1}{2} \sum_{t=1}^{T} [\zeta_{it} + \zeta_{jt}]$, where $\zeta_{it}$ denotes the normalized Copeland score of querying arm $i$ at round $t$.

### 2.3.2 LEARNING PREFERENCE WITH ACTIVE DUELING BANDIT

As the name suggests (Algorithm 1), our algorithm is a dueling-bandit-inspired active learning algorithm. Active learning has two core compositions (Settles, 2009; Ren et al., 2021): scenario and query strategy. Among the three scenarios (query synthesis, streamed-based selective sampling, and

pool-based), our research problem is most suitable for the pool-based active learning scenario since we have sufficient computation resources and select the next querying object based on the distribution of the entire collection. From the perspective of query strategy, active learning can be classified into uncertainty sampling, query-by-committee, and expected model change. Uncertainty sampling is most appropriate for our proposed method because we aim to provide a certainty level for recommendations, calculated in the uncertainty sampling.

In the case we have 5 arms (Fig. 2), after each round, we obtain a $5 \times 5$ predicted winning probability using equation (4). Subsequently, we calculate a utility matrix, denoting the weighted loss of each arm based on equation (5). The first candidate arm $c$ is selected if its row has the maximum cumulative loss. The second candidate arm $d$ is selected if, in the row corresponding to $c$, arm $d$ contributes the most to the cumulative loss. The essence of our proposed method lies in selecting the least certain solutions from the perspective of weighted loss function. After several iterations, our method gradually converges to an accurate prediction of the winning probability.

Our proposed method has several input parameters. $\kappa$ is a parameter controlling the trade-off between random search and greedy search. If $\kappa = 0$, then the sampling probability $u_{ij} = \frac{1}{K}$ for each arm, meaning we randomly choose the two comparison arms. The closer $\kappa$ is to 0, the more our algorithm explores. $T$ is the maximum iterative round. $B$ denotes the total budget, indicating the maximum number of times the dueling bandit algorithm is allowed to ask for comparison results from Oracle $\mathcal{O}$. Also, we assume that if the same question has been asked to $\mathcal{O}$, the result of consultation can be reused without consuming $B$.

Each round, we maintain two matrices. $\hat{\mathbf{P}} = [\hat{p}_{ij}]_{K \times K}$ is the predicted winning probability matrix:

$$\hat{\mathbf{P}} = \frac{\mathbf{w}}{\mathbf{w} + \mathbf{w}^\top} \tag{4}$$

where $\mathbf{w} = [w_{ij}]_{K \times K}$ stores the comparison results, $w_{ij}$ denotes the total number of times arm $i$ beats $j$. The denominator denotes the matrix storing the total comparison times between each pair of arms.

Also, our algorithm maintains a utility matrix, $\mathbf{U} = [u_{ij}]_{K \times K}$, which is used to measure the prediction accuracy for each arm. $\mathbf{U}$ is defined as follows:

$$\mathbf{U} = p_{\min} + (1 - Kp_{\min}) \frac{L(\hat{p}_{ij})}{\sum_{a_i, a_j \in \mathcal{A}} L(\hat{p}_{ij})} \tag{5}$$

where $p_{\min} = 1/(Kt^\kappa)$ is the trade-off minimum probability controlled by $\kappa$ (line 5 Algorithm 1). It's worth noticing that the loss function we use here is the mean square error (MSE):

$$L_{MSE}(\hat{\mathbf{P}}) = \frac{1}{K} \sum_{i=1}^{K} \sum_{j=1}^{K} (\hat{p}_{ij} - p_{ij})^2 \tag{6}$$

There are many other popular loss functions, such as logistic loss, squared loss and exponential loss (Ganti & Gray, 2012), which is most suitable for classification problems because they use traditional one-hot vecotr to indicate class.

Our proposed method has regret satisfying the following proposition.

**Proposition 1.** For any $t \in [T]$, if *RUCB-AL* runs with $\gamma = \frac{1}{t^\kappa} = K$, then the expected regret of Algorithm 1 satisfies (proof in Appendix A.2):

$$\mathbb{E}[R_T] \leq \frac{K^2 - K - 4}{K - 1} T + \log K$$

## 2.4 PREFERENCE ELICITATION MODULE

There are two critical metrics for every algorithm structure. The first, denoted as $c_1$, representing the first constraint, determining Z the first consultation should occur:

$$c_1 : current\_generation \geq \delta_1 \tag{7}$$

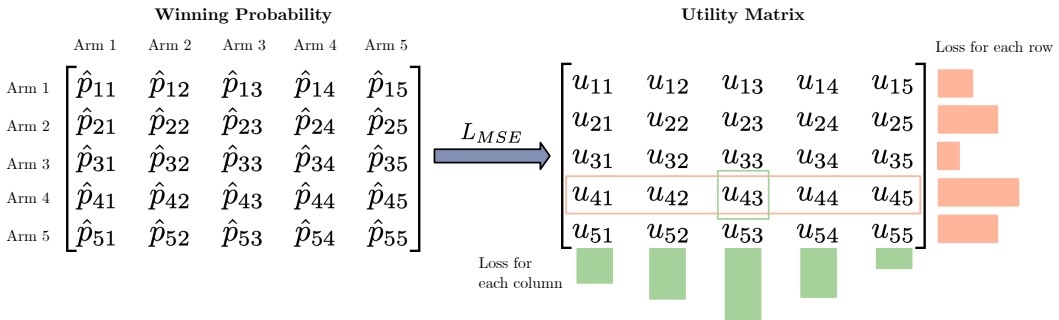

Figure 2: A deception of how RUCB-AL choose comparison pairs ($K = 5$)

where $\delta_1 = b \times G$ is the threshold for the first constraint, $b$ is the budget parameter for the first consultation, and $G$ denotes the maximum number of generations. In the experiment session, we assume when the evaluated generation reaches the pre-designed budget (e.g., $b = 0.4$), the first consultation can take place.

The second constraint, $c_2$, calculates the information gain $D_{KL}$ of two adjacent recommendations:

$$c_2 : D_{KL}(V_{s-1}, V_s) \geq \delta_2 \tag{8}$$

where $D_{KL}(V_{s-1}, V_s) = \sum_{\mathbf{z_i} \in Z} V_{s-1}(\mathbf{z_i}) \frac{\log(V_{s-1}(\mathbf{z_i}))}{\log(V_s(\mathbf{z_i}))}$, and $V_s$ is the virtual utility function denoting the predicted preference distribution of the current population at consultation session $s$, defined as follows:

$$V_s = \begin{cases} \mathcal{N}(\mathbf{z}_0^*, \sigma), & s = 0 \\ v_s(\mathbf{z}_s^*) + \lambda V_{s-1}, & \text{otherwise} \end{cases} \tag{9}$$

where $\lambda$ is the discount rate, $\mathbf{z}_s^*$ is the recommended solution in consultation session $s$. $V_s$ is only an assumption of the preference distribution which only cares about the gloval optima. $c_2$ calculates the difference between two distributions. When the recommendation of consultation module becomes stable, the predicted preference distribution will be almost the same, and thus $D_{KL}$ will have a small value (set $\delta_2 = e^{-3}$). When $D_{KL}$ reaches the small threshold value $\delta_2$, it is assumed that there is no need for further consultation. The structural PBEMO algorithms are listed below (the step-by-step process is available in Appendix A.3. ):

---

**Algorithm 2** Single-ojbective PBEMO

---

**Input:** max number of round $T$, $N$ number of pairwise comparisons.
  1: Uniformly sample $N$ pairwise comparisons as our dataset $\mathcal{D} = \{[\mathbf{z}_i, \mathbf{z}_i'], y_i\}_{i=1}^N$, where $y_i = 1$ denotes $\mathbf{z}_i \succ \mathbf{z}_i'$.
  2: run **RUCB-AL** with input $T$ and $\mathcal{D}$.
**Output:** The global optima $\mathbf{z}^*$.

---

**Algorithm 3** Dominance-Based PBEMO

---

**Input:** $G$ max number of generation, $N$ population number, $s$ consultation session.
  1: $s \leftarrow 0$
  2: **while** $current\_generation < G$ **do**
  3:   **if** $c_1$ is true and $c_2$ is true **then**
  4:     Update $V_s$ and select 10 incumbent solutions $\mathbf{z}_i, i = \{1, 2, \ldots, 10\}$ from current population according to $V_s$.
  5:     Feed $\mathbf{z}_i$ to **RUCB-AL**, and record recommendation $\mathbf{z}_s^*$.
  6:     Run NSGA-II by assigning fitness value with virtual fitness function.
  7:     $s \leftarrow s + 1$.
  8:   **else**
  9:     Run NSGA-II.
 10:   **end if**
 11: **end while**
**Output:** Population $\mathbf{z}_i, i \in \{1, 2, \ldots, N\}$.

---

---

**Algorithm 4** Decomposition-Based PBEMO

---

**Input:** $G$ max number of generation, $N$ population number, $s$ consultation session, $W = \{\mathbf{w}^i\}_{i=1}^N$ uniformly distributed weight vecotrs, $\mu$ number of best weight vecotor, $\eta$ step size.
1:   $s \leftarrow 0$
2:   **while** $current\_generation < G$ **do**
3:     **if** $c_1$ is true and $c_2$ is true **then**
4:       Update $V_s$ and select 10 incumbent solutions $\mathbf{z}_i, i = \{1, 2, \ldots, 10\}$ from current population according to $V_s$.
5:       Feed $\mathbf{z}_i$ to **RUCB-AL**, and record recommendation $\mathbf{z}_s^*$.
6:       Run NSGA-II by assigning fitness value with virtual fitness function.
7:       Select $\mu$ best points and store their corresponding weight vectors $W^V = \{\mathbf{w}^i\}_{i=1}^\mu$.
8:       Move the remainning reference points towards $\mathbf{w}^{V^i}$ as follows and collect new wegiht vectors $W'$:
$$w_j = w_j + \eta \times (w^{V^i} - w_j), (i = 1, 2, \ldots, \mu)$$

9:       $W \leftarrow W'$
10:      Run MOEA/D with new $W$.
11:      $s \leftarrow s + 1$.
12:     **else**
13:      Run MOEA/D.
14:     **end if**
15: **end while**
**Output:** Population $\mathbf{z}_i, i \in \{1, 2, \ldots, N\}$.

---

**Algorithm 5** Indicator-Based PBEMO

---

**Input:** $G$ max number of generation, $N$ population number, $s$ consultation session, $W = \{\mathbf{w}^i\}_{i=1}^N$ uniformly distributed weight vecotrs, $\mu$ number of best weight vecotor.
1:   $s \leftarrow 0$
2:   **while** $current\_generation < G$ **do**
3:     **if** $c_1$ is true and $c_2$ is true **then**
4:       Run the same as Algorithm 4 line 3-8.
5:       Recalculate R2 indicator (equation (33)).
6:       Run R2-IBEA with new $W$.
7:       $s \leftarrow s + 1$.
8:     **else**
9:      Run R2-IBEA.
10:    **end if**
11: **end while**
**Output:** Population $\mathbf{z}_i, i \in \{1, 2, \ldots, N\}$.

---

## 3   Empirical Study

### 3.1   Research Questions

In this paper, we mainly focus on the following 3 research questions (RQs):

- *RQ1*: What is the effect of different budget on active dueling bandit?
- *RQ2*: What is the performance of our proposed method on traditional test problem suite comparing to peer algorithms?
- *RQ3*: Whether our proposed method still works well on the practical problem?

The performance metrics are listed in Appendix A.4.

### 3.2   Effects of Budget on Active Dueling Bandit

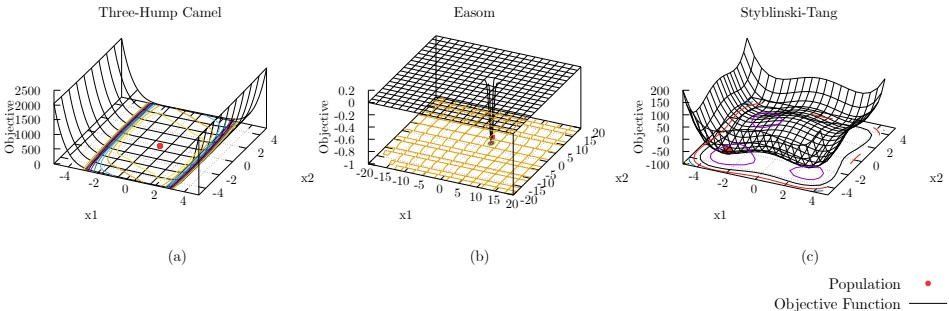

Figure 4: RUCB-AL running on six different single objective functions: (a) Sphere function, (b) Booth function, (c) Ackley, (d) Three-hump camel function, (e) Easom function, and (f) Styblinski-Tang function.

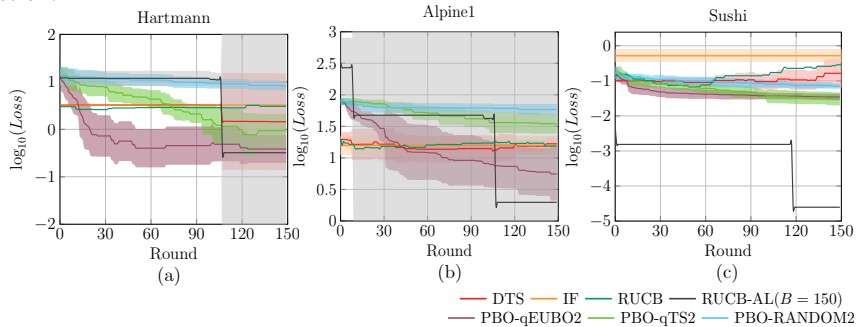

Figure 5: Comparing RUCB-AL with peer algorithms (e.g., DTS, IF, RUCB, PBO-qEUBO, PBO-qTS, PBO-random)

We evaluate the active learning ability of our proposed method in two sessions, on a toy problem and 9 black-box optimization problems (BBOPs).

**Toy Problems** We set the number of arms $K = 10$ with the optimal arm as $a^* = 1$, and the performance metric equation (6) guides our assessment. Real-world consultation results are expensive, so we limit the number of different queries to $B = \{20, 30, 40\}$ (Fig. 3 (a)) and compare it with the baseline algorithm, RUCB (Fig. 3 (b)). In Fig. 3 (a), a sharp increase occurs when the number of round is less than 10 due to the small number of comparisons. The loss stabilizes after a certain round due to the budget limitation. A larger budget allows for more rounds to be utilized.

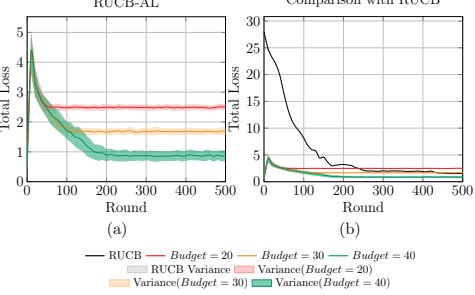

Figure 3: The active learning ability of RUCB-AL, (a) the total distance of RUCB-AL ($B = \{20, 30, 40\}$), (b) the comparison with baseline algorithm, RUCB

**BBOPs** We select Sphere, Booth, Ackley, Three-Hump Camel, Easom and Styblinski-Tang, Hartmann, Alpine1, and a 4-dimensional real over 100 sushi items (Sushi) (Kamishima, 2003). Peer algorithms are 3 traditional dueling bandit algorithms and 3 Bayesian optimization methods (i.e, qTS (Siivola et al., 2021), qEUBO (Astudillo et al., 2023), and random PBO). At each round the search space is 100 scatter points randomly and uniformly distributed in the feasible region. We set $B = 150$ for RUCB-AL and run repetitively 20 times. In our context, user preference is to find the global minimum value of the objective function $\mathbf{z}^r = -Inf$. Experiments result in Fig. 5, Fig. 8.

**Response to _RQ1_**: This subsection presents the results of an empirical study assessing the active learning ability of RUCB-AL under toy problems with 10 arms and budgets of $B = \{20, 30, 40\}$, as well as synthetic problems with 100 arms and a budget of 150. The findings indicate that our proposed method converges faster than the baseline algorithm with limited consultation. Fur-

thermore, with 10 arms, our method consistently outperforms RUCB when $B \geq 30$. In synthetic problems, RUCB-AL surpasses other peer algorithms except in cases where the objective function exhibits steep ridges.

### 3.3 PERFORMANCE OF OUR PROPOSED METHOD ON TEST PROBLEM SUITE

In the session, we implement the three different categorical PBEMO algorithms on ZDT ($m = 2$), DTLZ ($m = \{3, 5, 8, 10\}$) and WFG ($m = 3$) benchmark problems, specifically ZDT1~ZDT4, ZDT6, DTZL1~DTLZ4, WFG1, WFG3, WFG5, and WFG7 which exhibit various PF shapes. Additionally we choose six peer algorithms (i.e., I-MOEAD-PLVF (Li et al., 2019), I-NSGA2/LTR, I-MOEA/D/LTR, I-R2-IBEA/LTR (Li et al., 2023), IEMO/D (Tomczyk & Kadziński, 2019), I-MOEA/D-PPL (Huang & Li, 2023)). The number of decision variables and evaluation times are set as recommended, and runs 20 times repetitively. Our proposed method is limited to querying the consultation module for at most 10 times. In each consultation session, a maximum of 10 solutions are sampled. With the findings from RQ1, we set $B = 40$. The specific parameter settings and population results of our proposed method are detailed in Appendix A.6.

**Response to *RQ2***: Our proposed method achieves the minimum mean in 8 instances (Table 1), ranking second overall. However, from the perspective of the Wilcoxon signed-rank test (Wilcoxon, 1992), our method consistently outperforms the other algorithms most of the time. When applied to MOP test suites, our method demonstrates more stable and accurate performance with a limited query budget.

### 3.4 PERFORMANCE OF OUR PROPOSED METHOD ON PRACTICAL PROBLEM: PSP

In this section, five different structural proteins are selected (1K36, 1ZDD, 2M7T, 3P7K and 3V1A) to construct PSP problem as a multi-objective problem (Zhang et al., 2023). Four energy functions (i.e., Bound, dDFIRE, Rosetta, RWplus) serves as objective functions. Each protein structure is tested three times. Following the results of RQ2, we choose MOEA/D-RUCB-AL as our method. For simplicity, I-MOEA/D-PLVF, I-NSGA2/LTR and IEMO/D are selected as peer algorithms, as they are often favorable in RQ2.

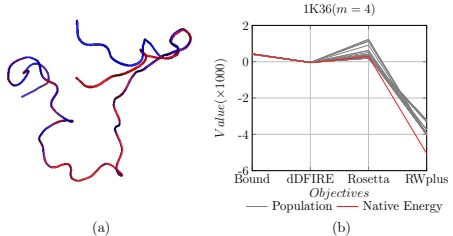

Fig. 6 (a) displays the molecular structure of native protein 1K36 alongside our predicted structure, red part denoting the native protein structure while blue for the predicted structure. Fig. 6 (b) illustrates the corresponding population arrangement of the predicted protein. The complete PSP experiments are available in Appendix A.7.

Figure 6: Running MOEA/D-RUCB-AL on PSP problems, for example protein 1K36, (a) the red color is the native protein structure and the blue color is our predicted protein structure, (b) the objective value of our predicted protein and the native protein

**Response to *RQ3***: Our proposed method demonstrates applicability to more sophisticated real-world problems and exhibits superior performance. As highlighted in RQ1, our method excels in real-world scenarios, potentially attributed to its utilization of **direct PBEMO**.

## 4 CONCLUSION

This paper introduces a direct PEBMO structure incorporating **RUCB-AL**, an active dueling bandit algorithm. The structure is versatile, addressing both SOPs and MOPs. Empirical studies demonstrate the active learning prowess of *RUCB-AL*. In benchmark problems, it outperforms peer algorithms with a more stable and accurate performance. Additionally, its application to a real-world problem, PSP, highlights its efficacy in complex scenarios compared to fitness-based preference learning algorithms.

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

# A    APPENDIX

## A.1    RELATED WORK

### A.1.1    PREFERENCE LEARNING

Preference learning can be divided into three categories: priori, posteriori and interactive based on the timing of consultation. Interactive preference elicitation (Li et al., 2020b) presents a valuable opportunity for the DM to gradually comprehend the underlying black-box system and consequently refine user preference information.

Preference learning has received significant attention in research. The process of preference learning can be categorized into two types: fitness-function based or fitness-function free. The first type, which is widely explored (Jacquet-Lagreze & Siskos, 1982; Fürnkranz & Hüllermeier, 2003; Chu & Ghahramani, 2005; Houlsby et al., 2011; 2012; Zintgraf et al., 2018), involves approximating a value function that represents user preference using mathematical tools such as Gaussian process (GP), neural network (NN) and others. Fitness-based method can be traced back to 1982. Jacquet-Lagreze & Siskos (1982) introduced the UTA (UTilités Additives) method for deducing value functions based on a provided ranking of reference set. In 2003, Fürnkranz & Hüllermeier (2003) employed pairwise preference to predict a ranking, representing a total order, for potential labels associated with new training examples. In 2005, Chu & Ghahramani (2005) utilized GP for pairwise preference learning (PGP) within a Bayesian framework. In 2007, Cao et al. (2007) proposed ListNet, a NN-based learning-to-rank method. In 2011, Houlsby et al. (2011) used GP classification integrated with information theory to select the best solution with maximum entropy. In 2012, Houlsby et al. (2012) extended their work to handle multi-user scenarios by introducing weight vector for each user and combining multiple preference latent functions. A study conducted in 2018 (Zintgraf et al., 2018) compared four prominent preference elicitation modules and found that ranking queries outperformed the pairwise and clustering approaches in terms of utility models ad human preference. However, recent studies have explored an alternative method of preference learning that do not involve approximating value functions. This approach is inspired by Yan et al. (2022) in 2022 who proposed a novel method for conducting pairwise preference judgments using double Thompson sampling (DTS) approach (Wu & Liu, 2016). Similar idea have also emerged in the realm of RL preference learning (Myers et al., 2023).

### A.1.2    PREFERENCE-BASED ACTIVE LEARNING

Preference-based active learning has been considered in the study by (Myers et al., 2023) and has been utilized in various fields such as classification tasks (Chen et al., 2013; 2017) and ranking aggregation (Chen et al., 2013). However none of these approaches address our specific problem of adaptively requesting pairwise comparisons between solutions in an online setting for PL. (Myers et al., 2023) also considers preference-based active learning (Settles, 2009; Ren et al., 2021; Kumar & Gupta, 2020). Outside RL, preference-based active learning has been used for classification tasks (Chen et al., 2013; 2017) and ranking aggregation (Chen et al., 2013). None of these approaches have tackled our problem of adaptively asking for pairwise comparisons between solutions at deployment to conduct preference learning in an online setting.

As mentioned earlier (Yan et al., 2022), the dueling bandit algorithm shows promise as a solution that does not involve the direct calculation of the value function. However, there is currently no dueling bandit algorithm designed specifically for active learning (Settles, 2009; Ren et al., 2021; Kumar & Gupta, 2020). The state-of-art active bandit algorithms have predominantly been developed for multi-armed bandit (MAB) problems (Baram et al., 2004; Antos et al., 2008; Carpentier et al., 2011; Ganti & Gray, 2012; 2013; Glimsdal & Granmo, 2019; Zhang et al., 2020). Baram et al. (2004) considered each arm as an active learner and proposed a method that used EXP3 and EXP4 to recommend the appropriate active algorithm for different scenarios. Antos et al. (2008) introduced GAFS-MAX, which selected under-sampled points or arms with maximum loss, and redefined the regret equation based on the loss function in active learning. Carpentier et al. (2011), using the same regret definition as Antos et al. (2008), constructed the upper confidence bound (UCB) function in two forms with the Chernoff-Heoffding and Bernstein allocation strategies respectively. Inspired by previous work (Ganti & Gray, 2012), Ganti & Gray (2013) proposed LCB-AL which aimed to minimize the uncertainty level measured by lower confidence bound (LCB).

### A.1.3 DUELING BANDIT

To address the problem of adaptively collecting pairwise comparisons between solutions in an online setting for preference learning, we propose an active dueling bandit algorithm. This algorithm not only actively selects solutions to be queried but also performs preference learning using pairwise comparisons, similar to the dueling bandit framework.

The dueling bandits problem involves a sequential decision-making process where a learner selects two out of $K$ "arms" in each round and receives real-valued feedback. As described in Bengs et al. (2021), dueling bandits can be grouped into 3 categories: MAB-related, merge sort/quick sort, and tournament/challenge. In this paper, our focus is on traditional dueling bandits, specifically those that fall within the MAB-related categorized. Among the traditional dueling bandit algorithms, there are four distinct methods for making pairwise comparisons. The first method is known as explore then commit (ETC), which is utilized by algorithms such as interleaved filtering (IF) (Yue et al., 2012), beat the mean (BTM) (Yue & Joachims, 2011) and SAVAGE (Urvoy et al., 2013). ETC methods kick out solutions that are unlikely to win, but this approach may lead to lower predictive probability accuracy. The second method involves using the upper confidence bound (UCB), for example relative upper confidence bound (RUCB) (Zoghi et al., 2014a), MergeRUCB (Zoghi et al., 2015), and relative confidence sampling (RCS) (Zoghi et al., 2014b). MergeRUCB, an extension of RUCB, is particularly designed for scenarios with a large number of arms. RCS combines UCB and Beta posterior distribution to recommend one arm for each duel in each iteration step. The third method employs Thompson sampling, as demonstrated by double Thompson sampling (DTS) (Wu & Liu, 2016) and MergeDTS (Li et al., 2020a). Similar to MergeRUCB, MergeDTS is designed for dealing with a substantial number of arms. It is worth nothing that UCB methods assume the existence of a Condorcet winner, whereas Thompson sampling methods assume a Copeland winner, representing a fundamental distinction between these two types. The fourth method involves using the minimum empirical divergence, as introduced by relative minimum empirical divergence (RMED) (Komiyama et al., 2015) and deterministic minimum empirical divergence (DMED) (Honda & Takemura, 2010). RMED and DMED employee KL divergence as a metric to evaluate candidate arms. Overall, these four methods represent different approaches to pairwise comparison in traditional dueling bandit algorithms. In this paper, our proposed method is inspired by RUCB. We build upon the assumption of the existence of a Copeland winner, strengthening the reliability of our proposed method. Additionally, similar to RUCB, our method incorporates the confidence level into the decision-making process. This feature allows us to provide a recommendation confidence.

### A.2 PROOF OF LEMMAS

**Proposition 1.** For any $t \in [T]$, if *RUCB-AL* runs with $\gamma = \frac{1}{t^\kappa} = K$, then the expected regret of Algorithm 1 satisfies:

$$\mathbb{E}[R_T] \leq \frac{K^2 - K - 4}{K - 1}T + \log K \tag{10}$$

The proof of the expected regret builds upon the following lemmas. We first bound magnitude of the estimates $p_{a_c}$, using the fact that $0 \leq \tilde{p}_{a_c} \leq 1$ where $\tilde{p}_{a_c}(i) = \frac{\sum_i L(\bar{p}_{ci})}{\sum_i \sum_j L(\bar{p}_{ij})}$.

**Lemma 1.** For all $t \in [T]$ and $i, j \in [K]$ it holds that $\frac{\gamma}{K} \leq p_{a_c} \leq 1 - \gamma + \frac{\gamma}{K}$, given $\gamma = \frac{1}{t^\kappa}$.

**Proof for Lemma 1.** According to the definition of $p_{a_c}$, we have $p_{a_c} = \frac{\gamma}{K} + (1 - \gamma)\tilde{p}_{a_c}$. So:

$$0 \leq \tilde{p}_{a_c} = \frac{(p_{ac} - \frac{\gamma}{K})}{1 - \gamma} \leq 1 \tag{11}$$

Let $\mathcal{H}_{t-1} := (x_{i1}, x_{j1}, q_1, \dots)$ denotes the history up to time $t$. We compute the expected instantaneous regret at time $t$ as a function of the Copeland scores at time $t$.

**Lemma 2.** For all $t \in [T]$ it holds that $\mathbb{E}[\mathbb{E}_{i \sim p(a_c)}[\zeta_{it}|\mathcal{H}_{t-1}]] = \mathbb{E}[p_{a_c}^\top \zeta_t]$.

**Proof for Lemma 2.**

$$\mathbb{E}[\mathbb{E}_{i \sim p(a_c)}[\zeta_{it}|\mathcal{H}_{t-1}]] = \mathbb{E}[\sum_{i=1}^{K} p_{a_c}(i)\zeta_t(i)] = \mathbb{E}[p_{a_c}^\top \zeta_t] \tag{12}$$

Then we bound the magnitude of the estimates $\zeta_t(i)$.

**Lemma 3.** For all $t \in [T]$ it holds that $\mathbb{E}[\zeta_t] \leq \frac{2}{K-1}\mathbb{E}[\bar{p}_{ij}]$.

**Proof for Lemma 3.** We have the quality of indicator function $\mathbb{E}[\mathbb{I}_A] = \int_X \mathbb{I}_A(x)dP = \int_A dP = P(A)$, so the left part can be processed:

$$\mathbb{E}[\zeta_t] = \frac{1}{K-1}\mathbb{E}[\mathbb{I}\{\bar{p}_{ij} > 1/2\}] \tag{13}$$

$$= \frac{1}{K-1}(1 \times P(\bar{p}_{ij} > 1/2) + 0 \times P(\bar{p}_{ij} < 1/2)) \tag{14}$$

$$= \frac{1}{K-1}P(\bar{p}_{ij} > 1/2) \tag{15}$$

Given Markov inequality, $P(X \geq a) \leq \frac{\mathbb{E}[X]}{a}$, we can further process the inequality:

$$\frac{1}{K-1}P(\bar{p}_{ij} > 1/2) \leq \frac{1}{K-1}\frac{\mathbb{E}[\bar{p}_{ij}]}{1/2} \tag{16}$$

$$= \frac{2}{K-1}\mathbb{E}[\bar{p}_{ij}] \tag{17}$$

Finally, we bound the second moment of our estimates.

**Lemma 4.** For all $t \in [T]$ it holds that $\mathbb{E}[\sum_{i=1}^{K} p_{a_c}(i)\zeta_t^2(i)] \leq \frac{4(1-\gamma+\frac{\gamma}{K})}{(K-1)^2}$

**Proof for Lemma 4.**

$$\mathbb{E}[\sum_{i=1}^{K} p_{a_c}(i)\zeta_t^2(i)] = \mathbb{E}[\sum_{i=1}^{K} p_{a_c}(i)\mathbb{E}[\frac{1}{K-1}\mathbb{E}[\mathbb{I}\{\bar{p}_{ij} > 1/2\}]]^2] \tag{18}$$

$$= \frac{1}{(K-1)^2}\mathbb{E}[\sum_{i=1}^{K} p_{a_c}(i)\mathbb{E}[\mathbb{I}\{\bar{p}_{ij} > 1/2\}]\mathbb{E}[\mathbb{I}\{\bar{p}_{ij} > 1/2\}]] \tag{19}$$

$$\leq \frac{1}{(K-1)^2}\mathbb{E}[\sum_{i=1}^{K} p_{a_c}(i)\sum_{j=1}^{K}(2\bar{p}_{ij})^2] \tag{20}$$

$$= \frac{4}{(K-1)^2}\mathbb{E}[\sum_{i=1}^{K}\sum_{j=1}^{K} p_{a_c}(i)\bar{p}_{ij}^2] \tag{21}$$

$$\leq \frac{4(1-\gamma+\frac{\gamma}{K})}{(K-1)^2} \tag{22}$$

The first and second equation is the expansion of formulation according to definition. The third line is processed using Markov inequality. After neatening the formulation in line 4, we further scale the equality by Lemma 1 and $\bar{p}_{ij}^2 \leq 1$.

**Proof overview.** We upper bound $R_T$, and recall that $R_T := \sum_{t=1}^{T} r_t = \zeta^* T - \frac{1}{2}\sum_{t=1}^{T} \zeta_{it} + \zeta_{jt}$. Note that $\mathbb{E}_{\mathcal{H}_T}[\zeta_t(i) + \zeta_t(j)] = \mathbb{E}_{\mathcal{H}_{t-1}}[\mathbb{E}_{i \sim p(a_c)}[\zeta_{it}|\mathcal{H}_{t-1}]]$, since $x_i$ and $x_j$ are i.i.d. Further note that we can write:

$$\mathbb{E}[R_T] = \sum_{t=1}^{T} r_t = \zeta^* T - \frac{1}{2}\sum_{t=1}^{T}[\zeta_{it} + \zeta_{jt}]$$

$$= \max_{k \in [K]}[\sum_{t=1}^{T} \zeta_t(k) - \frac{1}{2}\sum_{t=1}^{T}[\zeta_{it} + \zeta_{jt}]] \tag{23}$$

where the last equality holds since we assume the $p_{ij}$ are chosen obliviously ans so $a^*$ does not depend on the learning algorithm. Thus we can rewrite:

$$\mathbb{E}[R_T] = \max_{k \in [K]}[\sum_{t=1}^{T} \zeta_t(k) - \sum_{t=1}^{T} \mathbb{E}_{\mathcal{H}_{t-1}}[\mathbb{E}_{i \sim p(a_c)}[\zeta_{it}|\mathcal{H}_{t-1}]]] \tag{24}$$

From the regret guarantee of standard *Multiplicative Weights* algorithm (Arora et al., 2012) over the completely observed fixed sequence of reward vectors $\zeta_1, \zeta_2, \ldots, \zeta_T$ we have for any $k \in [K]$:

$$\sum_{t=1}^{T} \zeta_t(k) - \sum_{t=1}^{T} [\tilde{p}_{a_c}^{\top} \zeta_t] \leq \log K + \sum_{t=1}^{T} \sum_{k=1}^{K} \tilde{p}_{a_c} \zeta_{ti}^2 \tag{25}$$

Note that $\tilde{p}_{a_c} = \frac{(p_{a_c} - \frac{\gamma}{K})}{1 - \gamma}$. Let $a^* = \arg\max_{k \in [K]} \sum_{t=1}^{T} \zeta_t(k)$. Taking expectation on both sides of the above inequality for $k = a^*$, we get:

$$(1 - \gamma) \sum_{t=1}^{T} \zeta_t(k) - \sum_{t=1}^{T} [p_{a_c}^{\top} \zeta_t] \leq \log K + \sum_{t=1}^{T} \sum_{k=1}^{K} p_{a_c} \zeta_{ti}^2 \tag{26}$$

which by applying Lemma 2, Lemma 3 and Lemma 4 and the fact that $\zeta_t(a^*) \leq 1$, $\gamma = K$, we have:

$$\mathbb{E}[R_T] \leq \gamma T - \frac{4T\gamma}{K(K-1)} + \frac{4T}{(K-1)^2} + \log K$$
$$\leq \gamma T + (1 - \gamma) \frac{4T}{(K-1)^2} + \log K \tag{27}$$
$$\leq \frac{K^2 - K - 4}{K - 1} T + \log K$$

### A.3 THE STEP OF FOUR DIFFERENT ALGORITHM ARCHITECTURE

#### A.3.1 DOMINANCE-BASED EMO ALGORITHM

This section will discuss how the learned preference information can be used in dominance-based EMO algorithms, e.g., NSGA-II (Deb et al., 2002b). Based on Deb & Sundar (2006), solutions from the best non-domination levels are chosen front-wise as before and a modified crowding distance operator is used to choose a subset of solutions from the last front which cannot be entirely chosen to maintain the population size of the next population, the following steps are performed:

Step 1: Before the first consultation session, the NSGA-II runs as usual without considering the preference information.

Step 2: If it is time to consult for the first time (e.g., when we have evaluated the population for $40\%$ of the total generation), then randomly selected 10 points fed into the consultation module and the best point $W^*$ recommended by *RUCB-AL* will be recorded and used to initialize the predicted preference distribution for the current population $L_0 = l_0(\mathbf{z}) = \mathcal{N}(W^*, \sigma)$.

Step 3: If the recommendation is not stable (e.g., the KL divergence between two adjacent predicted distributions is bigger than the threshold $\delta_2$), then points are sampled according to $U_{t-1}$ and the best point $W^*$ recommended by *RUCB-AL* is recorded and used to update the predicted distribution for the current population $l_t(\mathbf{z}) = \mathcal{N}(W^*, \sigma)$, $L_t = l_t + \lambda L_{t-1}$, where $t$ is the consultation count.

Step 4: Between two interactions, the crowding distance of each solution will be evaluated by the predicted preference distribution learned from the last consultation session.

#### A.3.2 DECOMPOSITION-BASED EMO ALGORITHM

Following Li et al. (2019), the decomposition-based EMO EMO algorithm (e.g., MOEA/D (Zhang & Li, 2007)) is designed to use a set of evenly distributed weight vectors $W = \mathbf{w}^i{}_{i=1}^N$ to approximate the whole PF. The recommendation point learned from the consultation module is to adjust the distribution of weight vectors. The following four-step process is to achieve this purpose.

Step 1: Before the first consultation session, the EMO algorithm runs as usual without considering any preference information.

Step 2: If time to consult for the first time (e.g., when we have evaluated the population for $40\%$ of the total generation), then randomly selected 10 points are fed into the consultation module and the best point $W^*$ recommended by *RUCB-AL* will be recorded and used to initialize the predicted preference distribution for the current population $L_0 = l_0(\mathbf{z}) = \mathcal{N}(W^*, \sigma)$. Select $\mu$ points closest to the reference point $\{W^{L^i}\}_{i=1}^{\mu}$.

Step 3: If the recommendation is not stable (e.g., the KL divergence between two adjacent predicted distributions is bigger than the threshold $\delta_2$), then points are sampled according to $L_{t-1}$ and the best point $W^*$ recommended by *RUCB-AL* is recorded and used to update the predicted distribution for the current population $u_t(\mathbf{z}) = \mathcal{N}(W^*, \sigma)$, $L_t = l_t + \lambda L_{t-1}$, where $t$ is the consultation count. Select $\mu$ points closest to the reference point $\{W^{L^i}\}_{i=1}^{\mu}$.

Step 4: Move the remaining reference points towards $\mathbf{w}^{U^i}$ as follows:

$$w_j = w_j + \eta \times (w^{L^i} - w_j), (i = 1, 2, \ldots, \mu) \tag{28}$$

Output the adjusted weight vectors as the new $W'$.

### A.3.3 INDICATOR-BASED EMO ALGORITHM

The R2 indicator was proposed to evaluate the relative quality of two sets of individuals (Hansen & Jaszkiewicz, 1994) from the standard weighted Tchebycheff function with a particular reference point $\mathbf{z}^r$ as follows:

$$R2(Z, W, \mathbf{z}^r) = \sum_{i=1}^{m} (p(w) \times \min_{\mathbf{z}_i \in Z} \{ \max_{1 \leq j \leq m} w_i |\mathbf{z}_j - \mathbf{z}_j^r| \}) \tag{29}$$

$W$ denotes a set of weight vectors. $p$ denotes a probability distribution on $W$. When the weight vectors are chosen uniformly distributed in the objective space, the R2 indicator is denoted as:

$$R2(Z, W, \mathbf{z}^r) = \frac{1}{|W|} \sum_{w \in W} (\min_{\mathbf{z}_i \in Z} \{ \max_{1 \leq j \leq m} w_i |\mathbf{z}_j - \mathbf{z}_j^r| \}) \tag{30}$$

where $\mathbf{z}^r$ is the ideal point.

R2-IBEA (Phan & Suzuki, 2013) performs parent selection and environmental selection with a binary R2 indicator:

$$I_{R2}(x, y) = R2(\{x\}, W, \mathbf{z}^*) - R2(\{x \cup y\}, W, \mathbf{z}^r) \tag{31}$$

$I_{R2}$ is designed to determine a superior-inferior relationship between given two individuals ($x$ and $y$) with two R2 values. If $x \succ y$, $I_{R2}(x, y) \geq 0$. In this case, we can get the property of weak monotonicity:

$$\begin{aligned} I_{R2}(x, y) \leq I_{R2}(y, x) \quad &\text{if } x \succ y \\ I_{R2}(x, y) \geq I_{R2}(y, x) \quad &\text{if } y \succ x \end{aligned} \tag{32}$$

In this section, we will use the recommended point $W^*$ from the consultation module to adjust the distribution of weight vectors in equation (33). The method of adjusting the distribution of weight vectors is the same as decomposition-based EMO algorithm. $W'$ is the adjusted weight vectors. The adjusted R2 indicator is denoted as:

$$R2'(Z, W, \mathbf{z}^*) = \frac{1}{|W|} \sum_{w \in W'} (\min_{\mathbf{z}_i \in Z} \{ \max_{1 \leq j \leq m} w_i |\mathbf{z}_j - \mathbf{z}_j^*| \}) \tag{33}$$

In this case, the interactive indicator-based EMO algorithm via progressively learned preference runs as following steps:

Step 1: Before the first consultation session, the R2-IBEA algorithm runs as usual without considering any preference information.

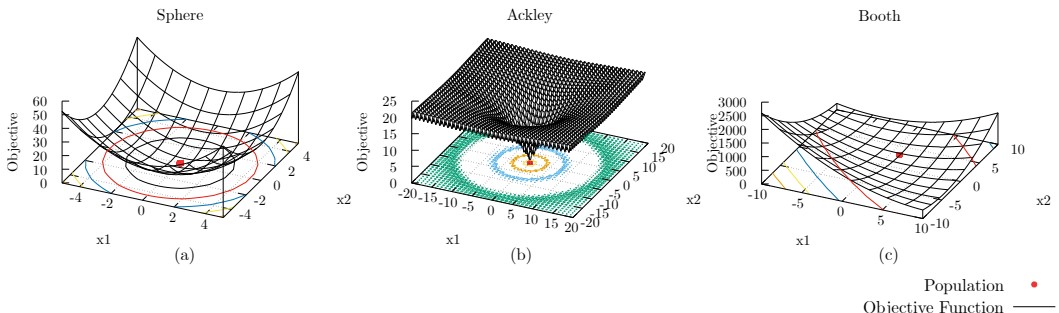

Figure 7: RUCB-AL running on six different single objective functions: (a) Sphere function, (b) Booth function, (c) Ackley, (d) Three-hump camel function, (e) Easom function, and (f) Styblinski-Tang function.

Step 2: If time to consult for the first time (e.g., when we have evaluated the population for $40\%$ of the total generation), then randomly selected 10 points are fed into the consultation module and the best point $W^*$ recommended by *RUCB-AL* will be recorded and used to initialize the predicted utility distribution for the current population $L_0 = l_0(\mathbf{z}) = \mathcal{N}(W^*, \sigma)$. Select $\mu$ points closest to the reference point $\{W^{L_i}\}_{i=1}^{\mu}$.

Step 3: If the recommendation is not stable (e.g., the KL divergence between two adjacent predicted distributions is bigger than the threshold $\delta_2$), then points are sampled according to $L$ and the best point $W^*$ recommended by *RUCB-AL* is recorded and used to update the predicted distribution for the current population $l_t(\mathbf{z}) = \mathcal{N}(W^*, \sigma)$, $L_t = l_t + \lambda L_{t-1}$, where $t$ is the current generation number. Select $\mu$ points closest to the reference point $\{W^{L^i}\}_{i=1}^{\mu}$.

Step 4: Adjust the distribution of weight vectors as the same as decomposition-based EMO algorithms, and recalculate the R2 indicator of every individual in populations.

## A.4 PERFORMANCE METRICS

For preference learning in SOPs, performance metric is the regret or the loss to optimal solution. Here, regret is distinct from dueling bandit regret and is defined as follows:

$$Loss = \mathbf{z}^r - z^* \tag{34}$$

where $\mathbf{z}^r$ denotes the optimal objective value and $\mathbf{z}^*$ is the recommended objective value corresponding to $W^*$.

Similarly, for MOPs, the performance metric is defined as:

$$Loss = \min_{\mathbf{z} \in Q} dist(\mathbf{z}, \mathbf{z}^r) \tag{35}$$

where $dist(\mathbf{z}, \mathbf{z}^r)$ is the Euclidean distance between $\mathbf{z}^r$ and a solution $\mathbf{z} \in Q$ in the objective space.

## A.5 EXPERIMENT ON SOP

The other three 2-dimensional BBOPs are listed in Fig. 7. The rest 6 BBOP problem comparison result are listed Fig. 8.

## A.6 EXPERIMENT ON TEST PROBLEM SUITE

### A.6.1 PARAMETER SETTING

This section lists several parameters that we need to set in advance, including the parameters in MOEA/D, the number of populations presented to DM and other parameters we need when combining DM with EAs:

- The probability and distribution of index for SBX: $p_c = 1.0$ and $\eta_c = 20$;

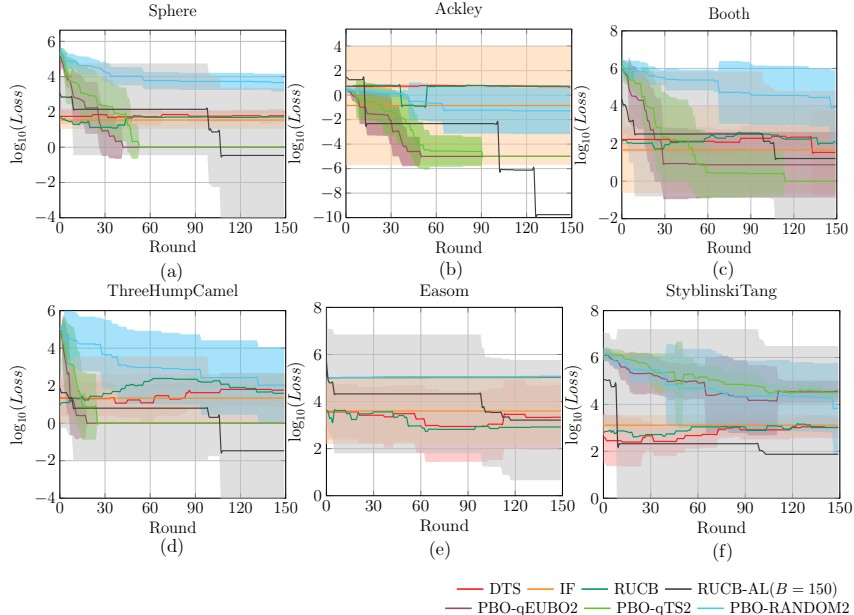

Figure 8: Comparing RUCB-AL with peer algorithms (e.g., DTS, IF, RUCB, PBO-qEUBO, PBO-qTS, PBO-random)

- The mutation probability and distribution of index for polynomial mutation operator: $p_m = \frac{1}{m}$ and $\eta_m = 20$;

- The maximum assessment number $N$ is set as $N = 25000$ for ZDT test suite while $N = 30000$ for DTLZ test suite;

- The population size is set as $pop\_num = 100$ for ZDT test suite and $pop\_num = 120$ for DTLZ test suite;

- The number of incumbent candidate presented to DM for consultation: $\mu = 10$;

- For fitness-function based PL algorithms, there exists the number of consecutive consultation session $\tau$ to be set. For all fitness-function based PL algorithms, we set: $\tau = 25$;

- The step size of reference point update $\eta$ for MOEA/D series algorithms are set to: $\eta = 0.3$.

### A.6.2 STATISTICAL TEST

To offer a statistical interpretation of the significance of comparison results, we conduct each experiment 20 times. To analyze the data, we employ the Wilcoxon signed-rank test (Wilcoxon, 1992) in our empirical study.

The Wilcoxon signed-rank test, a non-parametric statistical test, is utilized to assess the significance of our findings. The test is advantageous as it makes minimal assumptions about the data's underlying distribution. It has been widely recommended in empirical studies within the EA community (Derrac et al., 2011). In our experiment, we have set the significance level to $p = 0.05$.

### A.6.3 POPULATION RESULTS

In this section, we show the results of our proposed method running on ZDT, DTLZ, and WFG test suites. For simplicity, we only show MOEAD-RUCB-AL for it outperform the other two algorithm.

The population results of MOEA/D-RUCB-AL running on ZDT1~ZDT4, and ZDT6 are shown in Fig. 9. The population results running on DTLZ1~DTLZ2 ($m = \{3, 5, 8, 10\}$) are shown in Fig. 10 Fig. 11 Fig. 12 Fig. 13 respectively. The results running on WFG ($m = 3$) are shwon in Fig. 14.

Table 1: THE MEAN(STD) VALUE OF OUR PROPOSED METHOD AND PEER ALGORITHMS RUNNING ON BENCHMARK PROBLEMS

| Test Porblem | | RUCB-AL | | | PLVF | LTR | | | IEMO/D | PPL |
|---|---|---|---|---|---|---|---|---|---|---|
| | | NSGA2 | MOEA/D | R2-IBEA | | NSGA2 | MOEA/D | R2-IBEA | | |
| ZDT1 | $m=2$ | 7.76E-2(4.62E-2) | **3.91E-2(1.02E-4)** | 1.11E-2(1.83E-1) | 4.99E-2(2.51E-2)† | 4.04E-2(1.13-2)† | 3.26E-1(8.56E-2)† | 1.09E-1(1.02E-1)† | 4.30E-2(8.02E-3)† | 8.55E-2(3.50E-2)† |
| ZDT2 | $m=2$ | **1.45E-1(7.94R-3)** | 2.09E-1(7.11E-2) | 2.42E-1(6.47E-2) | 3.62E-1(1.57E-1)† | 1.46E-1(3.53E-2) | 8.96E-1(1.61E-2)† | 2.83E-1(6.81E-7)† | 2.20E-1(1.71E-1)† | 2.82E-1(3.37E-2)† |
| ZDT3 | $m=2$ | 1.23E-1(7.48E-2) | 1.78E-1(2.11E-2) | 1.77E-1(3.08E-1) | 3.72E-1(1.87E-1)† | **7.01E-2(5.17E-4)** | 1.15(7.82E-2)† | 1.91E-1(1.38E-1)† | 1.24E-1(2.61E-2)† | 1.17(1.08E-1)† |
| ZDT4 | $m=2$ | 8.01E-2(4.66E-2) | 8.04E-2(4.6E-2) | 1.43E-1(8.65E-2) | 7.52E-2(4.72E-2) | **4.34E-2(3.48E-2)** | 2.49E-1(1.10E-1)† | 9.45E-2(7.93E-2) | 7.28E-2(3.07E-2)† | 1.28E-1(7.68E-2) |
| ZDT6 | $m=2$ | 6.41E-2(3.97E-2) | 5.49E-2(4.61E-3) | 7.66E-2(4.59E-2) | 1.16E-1(8.19E-2)† | **3.85E-2(3.63E-3)‡** | 6.30E-1(1.24E-2)† | 2.21E-1(1.04E-1)† | 5.44E-2(3.43E-3) | 2.32E-1(1.70E-2)† |
| WFG1 | $m=3$ | 2.33(3.71E-2) | **1.11(2.08E-1)** | 2.39(1.68E-2) | 2.01(6.55E-2)† | 1.28(2.28E-16)† | 1.95(2.48E-2)† | 1.61(4.56E-16)† | 2.63(3.09E-1)† | 2.01(6.55E-2)† |
| WFG3 | $m=3$ | 7.10E-1(7.77E-2) | **6.45E-1(1.65E-2)** | 2.25(1.24) | 8.91E-1(2.53E-1)† | 1.11(1.31E-1)† | **6.45E-1(2.26E-2)** | 1.01(3.10E-2)† | 7.48E-1(1.27E-2)† | 8.91E-1(2.53E-1)† |
| WFG5 | $m=3$ | 3.49(1.92) | 3.01(6.42E-1) | 3.52(1.43) | 2.68(4.62E-1)† | 1.94(5.19E-3)‡ | **1.67(8.25E-4)‡** | 2.01(8.24E-2)† | 2.6(9.48E-2)† | 1.75(1.52E-2)† |
| WFG7 | $m=3$ | **1.29(4.70E-2)** | 3.24(3.89E-1) | 1.84(1.31) | 2.63(5.28E-1)† | 1.34(4.23E-2) | 1.30(2.93E-3) | 1.52(3.32E-2)† | 1.49(6.99E-2)† | 2.63(5.28E-1)† |
| DTLZ1 | $m=3$ | 2.79E-1(8.54E-2) | 1.72E-1(1.72E-3) | 2.95E-1(6.85E-2) | 1.63E-1(3.48E-2) | 1.94E-1(6.91E-2) | **1.44E-1(2.79E-2)** | 1.79E-1(1.16E-2) | 1.59E-1(1.69E-2)† | 3.25E-1(1.26E-1)† |
| | $m=5$ | 3.07(2.34) | 3.67E-1(7.36E-2) | 4.02E-1(8.65E-2) | 2.96E-1(1.25E-1)† | 3.57E-1(1.16E-1) | 2.00E-1(8.84E-2)† | 3.25E-1(2.78E-2)† | **1.96E-1(1.07E-1)‡** | 2.87E-1(1.27E-1)† |
| | $m=8$ | 1.51(1.17) | 5.68E-1(5.15E-2) | 5.57E-1(1.45E-2) | 3.07E-1(1.88E-1)‡ | 4.75E-1(2.79E-1) | 2.85E-1(2.07E-1)‡ | 5.10E-1(1.04E-2)‡ | **2.54E-1(2.36E-1)‡** | 3.92E-1(1.83E-1)‡ |
| | $m=10$ | 5.02(3.70) | 3.36E-1(5.12E-2) | 4.68E-1(5.91E-2) | 2.20E-1(8.15E-2)‡ | 5.63E-1(4.76E-1)† | 1.67E-1(6.63E-2)‡ | 2.94E-1(2.17E-2)‡ | **1.52E-1(7.28E-2)‡** | 2.31E-1(1.09E-1)‡ |
| DTLZ2 | $m=3$ | 3.40E-1(1.90E-1) | **1.82E-1(2.31E-2)** | 2.25E-1(7.34E-2) | 2.43E-1(5.18E-2)† | 1.86E-1(1.35E-2) | 2.07E-1(1.06E-2)† | 2.25E-1(7.34E-2)† | 1.85E-1(9.75E-3)† | 5.72E-1(1.55E-1)† |
| | $m=5$ | 5.91E-1(1.05E-1) | 4.66E-1(5.19E-2) | 1.21(1.25E-1) | 5.21E-1(1.47E-1) | 5.39E-1(1.67E-1) | 5.09E-1(1.67E-1)† | 4.95E-1(9.11E-2) | **3.64E-1(1.28E-2)‡** | 6.34E-1(8.20E-2)† |
| | $m=8$ | 1.33(1.35) | 8.06E-1(1.02E-1) | 1.30(3.27E-1) | 6.97E-1(1.92E-1) | 9.04E-1(2.18E-1) | 6.21E-1(1.48E-1)† | 7.46E-1(8.19E-2)† | **4.13E-1(1.63E-1)‡** | 1.01(2.49E-1)† |
| | $m=10$ | 1.22(1.07E-1) | 6.51E-1(2.02E-1) | 6.46E-1(1.20E-1) | 6.43E-1(1.78E-1) | 8.57E-1(1.44E-1)† | 4.76E-1(1.03E-1)† | 4.66E-1(1.32E-1)† | **3.39E-1(1.69E-1)‡** | 8.81E-2(6.96E-2)† |
| DTLZ3 | $m=3$ | 4.48(1.41) | 1.93(1.42) | 5.49(2.25) | **1.55(9.25E-1)** | 6.16(3.71)† | 2.76(2.46) | 3.42(1.17)† | 1.59(1.80) | 3.24(1.85)† |
| | $m=5$ | 6.18(5.04) | 4.88E-1(6.60E-2) | 1.14(5.36E-2) | 8.43E-1(3.80E-1)† | 1.71E+1(9.60)† | 1.05(5.90E-1)† | 5.66E-1(1.97E-1) | **3.73E-1(5.92E-2)‡** | 9.24E-1(1.78)† |
| | $m=8$ | 4.29E+1(1.57E+1) | 9.67E-1(1.85E-1) | 8.87E-1(3.62E-1) | 8.56E-1(1.82E-1) | 3.04E+1(1.46E+1)† | 6.48E-1(1.91E-1)† | 9.13E-1(1.25E-1) | **5.48E-1(2.29E-1)‡** | 1.13(2.75)† |
| | $m=10$ | 5.81E+1(2.07E+1) | 7.26E-1(1.55E-1) | 8.25E-1(1.82E-1) | 6.54E-1(1.69E-1) | 2.86E+1(1.31E+1)† | 4.31E-1(1.13E-1)‡ | 5.11E-1(1.04E-1)‡ | **2.95E-1(1.27E-1)‡** | 8.87E-1(1.04E-1)† |
| DTLZ4 | $m=3$ | 2.59E-1(8.15E-2) | **1.84E-1(1.25E-2)** | 2.50E-1(2.91E-1) | 5.73E-1(3.58E-1)† | 6.83E-1(3.75E-1)† | 6.11E-1(3.13E-1)† | 6.34E-1(2.50E-1)† | 6.00E-1(3.16E-1)† | 7.72E-1(2.47E-1)† |
| | $m=5$ | 6.71E-1(1.49E-1) | **5.07E-1(9.73E-2)** | 9.33E-1(2.65E-1) | 6.08E-1(1.42E-1)† | 1.02(1.61E-1)† | 6.99E-1(1.98E-1)† | 5.80E-1(1.34E-1) | 5.99E-1(1.84E-1)† | 6.37E-1(7.99E-2)† |
| | $m=8$ | **7.13E-1(1.46E-1)** | 8.79E-1(6.04E-2) | 9.64E-1(2.87E-1) | 7.88E-1(2.26E-1) | 1.01(1.15E-1)† | 8.90E-1(2.10E-1)† | 8.33E-1(1.12E-1)† | 7.18E-1(2.38E-1) | 1.12(2.61)† |
| | $m=10$ | 1.14(1.58E-1) | 6.87E-1(1.11E-1) | 4.23E-1(1.64E-1) | 7.03E-1(1.61E-1) | 1.27(1.31E-2)† | 7.48E-1(1.02E-1)† | **5.21E-1(1.04E-1)‡** | 5.75E-1(1.90E-1)† | 9.15E-1(6.14E-2)† |

† denotes our proposed method significantly outperforms other peer algorithms according to the Wilcoxon's rank sum test at a 0.05 significance level;

‡ denotes the corresponding peer algorithm outperforms our proposed algorithm.

Table 2: THE DIFFERENCE BETWEEN NATIVE AND PREDICTED PROTEIN IN ENERGY

| ID | Type | Bound | dDFIRE | Rosetta | RWplus |
|---|---|---|---|---|---|
| 1K36 | Native | 431.51 | -52.84 | 293.70 | -5059.39 |
| | Predicted | 431.75 | -41.66 | 402.33 | -3990.52 |
| 1ZDD | Native | 297.18 | -74.02 | -27.73 | -4604.18 |
| | Predicted | 328.84 | -63.03 | 63.03 | -3986.78 |
| 2M7T | Native | 269.76 | -39.51 | -10.82 | -3313.84 |
| | Predicted | 276.12 | -22.98 | 210.47 | -2111.19 |
| 3P7K | Native | 379.04 | -104.15 | -11.29 | -6140.81 |
| | Predicted | 413.47 | -91.21 | 184.17 | -3399.93 |

## A.7 EXPERIMENT ON PSP

In this section we listed the results implementing our proposed method, namely MOEA/D-RUCB-AL on PSP problems. We implement RMSD as the performance metric for PSP problems:

$$RMSD = \sqrt{\frac{\sum_{i=1}^{m} d_i^2}{m}} \tag{36}$$

where $d$ is the distance between each pair of atoms. The performance comparison results are in Table 2.

The red part in predicted protein structures( Fig. 15) represents the native protein and blue represent our predicted protein structure. The population results are shown in Fig. 16. The RMSD comparison results are shown in Table 3. As we can see our proposed method have better convergence and acuuracy than synthetic problems. This may be caused by two reasons:

- The first one is the PSP problem is only conducted on 4-dimensional objective spaces. In RQ2 our proposed method performs better in low dimensional problems.

- The second reason is the formulation of PSP problems. In this paper, we adopt utilizing 4 energy function to represent, which are empirically proved to be more accurate than in 1-dimensional objective function(Zhang et al., 2023).

Table 3: THE MEAN (STD) OF RMSD COMPARING OUR PROPOSED METHOD WITH PEER ALGORITHMS ON PSP PROBLEMS

| ID | MOEA/D-RUCB-AL | I-MOEA/D-PLVF | I-NSGA2-LTR | IEMO/D |
|---|---|---|---|---|
| 1K36 | **583.29(117.08)** | 682.23(182.63)† | 597.19(284.91) | 610.62(402.31)† |
| 1ZDD | **446.88(542.33)** | 623.14(394.14)† | 450.23(582.19) | 488.28(518.42)† |
| 2M7T | 350.51(8.95) | 671.45(372.01)† | 721.73(502.31)† | 823.46(1023.54)† |
| 3P7K | 719.90(1202.92) | **663.29(802.99)‡** | 692.31(823.13)‡ | 818.93(923.87) |
| 3V1A | **687.07(497.33)** | 887.68(391.74)† | 791.13(304.72)† | 823.28(528.87)† |

† denotes our proposed method significantly outperforms other peer algorithms according to the Wilcoxon's rank sum test at a 0.05 significance level;
‡ denotes the corresponding peer algorithm outperforms our proposed algorithm.

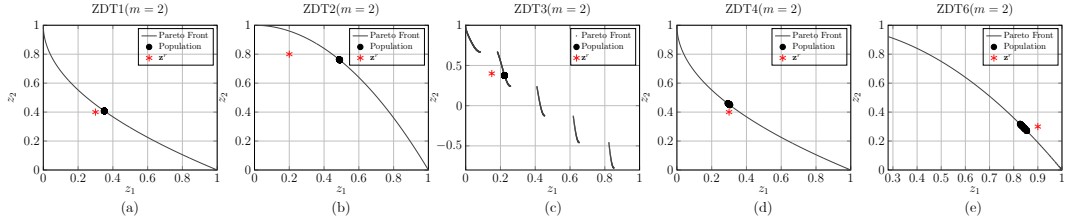

Figure 9: The population distribution of our proposed method (e.g., MOEA/D-RUCB-AL) running on ZDT test suite ($m = 2$)

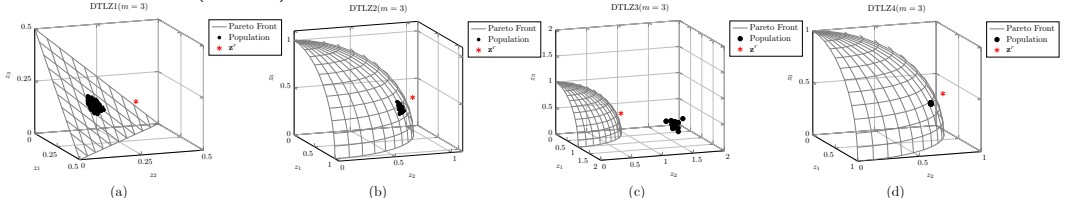

Figure 10: The population distribution of our proposed method (e.g., MOEA/D-RUCB-AL) running on DTLZ test suite ($m = 3$)

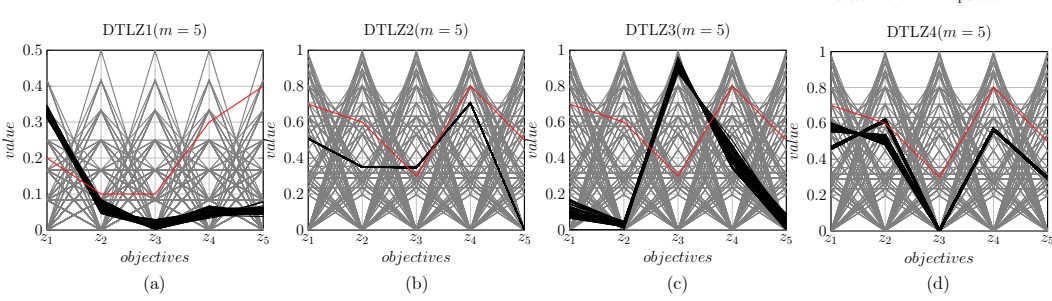

Figure 11: The population distribution of our proposed method (e.g., MOEA/D-RUCB-AL) running on DTLZ test suite ($m = 5$)

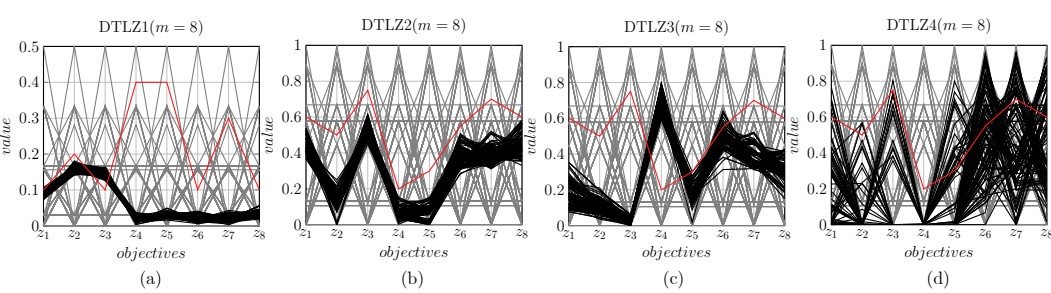

Figure 12: The population distribution of our proposed method (e.g., MOEA/D-RUCB-AL) running on DTLZ test suite ($m = 8$)

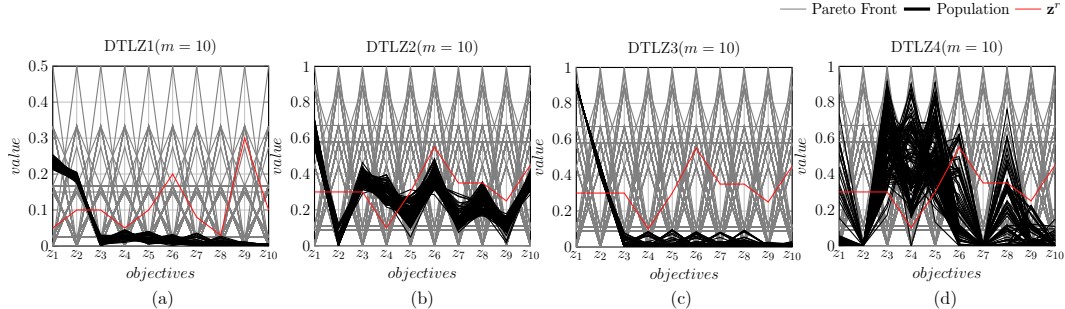

Figure 13: The population distribution of our proposed method (e.g., MOEA/D-RUCB-AL) running on DTLZ test suite ($m = 10$)

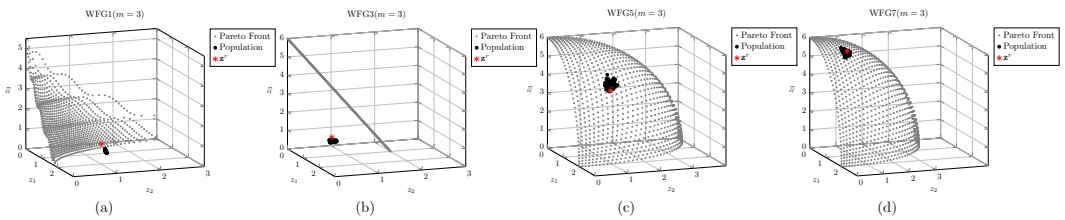

Figure 14: The population distribution of our proposed method (e.g., MOEA/D-RUCB-AL) running on WFG test suite ($m = 3$)

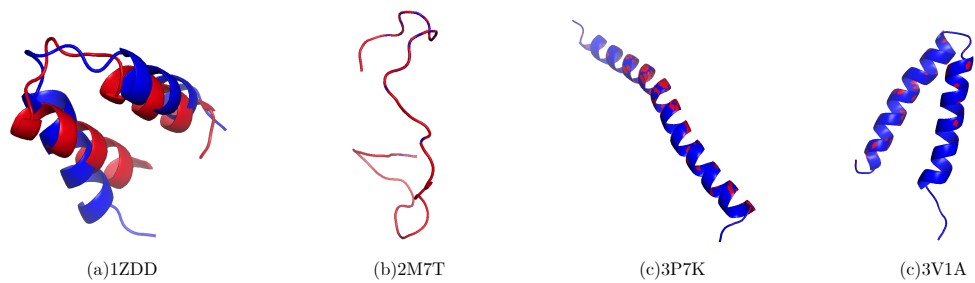

(a)1ZDD    (b)2M7T    (c)3P7K    (c)3V1A

Figure 15: The protein structure of our proposed method (e.g., MOEA/D-RUCB-AL) running on PSP problems ($m = 4$)

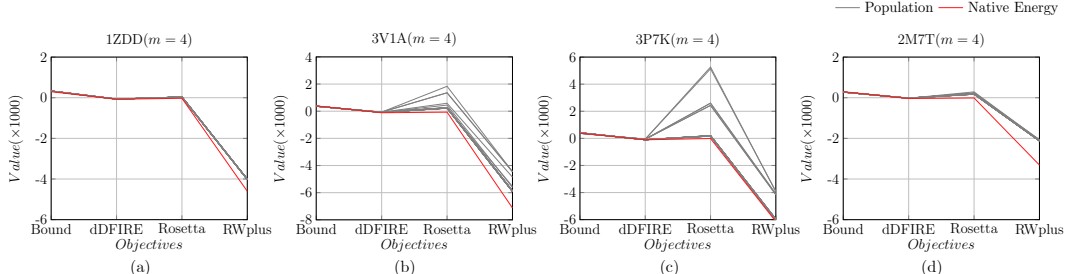

Figure 16: The population distribution of our proposed method (e.g., MOEA/D-RUCB-AL) running on PSP problems ($m = 4$)

