# OpenReview forum: "Efficient Interactive Preference Learning in Evolutionary Algorithms: Active Dueling Bandits and Active Learning Integration"
_ICLR.cc/2024/Conference — Submitted to ICLR 2024_

### Official Review · Reviewer_5t57 · 2023-10-27

**Soundness:** 2 fair
**Presentation:** 1 poor
**Contribution:** 1 poor
**Rating:** 3
**Confidence:** 3

**Summary:**

The authors propose an algorithm for learning based on pairwise preferences in a dueling bandit setting. The proposed method also includes a module for determining the query times, at which the user is queried for preference feedback. The algorithm is then evaluated empirical using single and multi-objective test functions and the sushi domain. It is also applied to a protein structure prediction problem.

**Strengths:**

Most preferential optimization approaches control the query times via batch-size hyperparameters or do not consider this problem at all. Therefore, this is an interesting addition. Furthermore, the number of experiments is quite high, spanning everything from very simple to real-world problems, including a diverse range of comparison algorithms.

**Weaknesses:**

Most importantly, the work lacks clarity:
- $\hat{P}$ denotes the predicted preference matrix, but there is no prediction step described. According to Eq. 4, it seems to be the observed preferences.
- Unclear how $P$, $\mathbf{w}$ and $\hat{P}$ are related.
- $P$ and $\hat{P}$ are formalized as probability matrix, but "user preferences are consistent and stationary", which is even used in the Algorithm 1 by reusing previous results. Resultingly $p_{ij}$ is an indictor function?
- Eq. 6 is defined over the matrix $\hat{P}$, but used with scalar values in Eq.5 ?
- Eq. 6 depends on $P$, but it is not clear how this is obtained.
- $L_t$ denotes the predicted preference distribution. How is this different from $\hat{P}$ and $\mathbf{w}$?
- $L$ is used as symbol for the loss as well as the preference distribution
- $p_min$ is never described and only defined within Algorithm 1
- Fitness is never introduced, but is likely to be assumes as a function of $\mathbf{F}(x)$. However, it seems to be also assumed that this function is linear, as Section 2.3 mentions "weight vectors"?
- Unclear how preferences are generated, but it does not seem to be possible to employ arbitrary generation processes, as a scalar fitness function is not able to result in cyclic preferences.
- Virtual fitness function not formally introduced

This also relates to how the bandit algorithm is embedded into the EA methods.
- It seems the bandit algorithm is used to evaluate candidate solutions, but in how far is the EA algorithm still working with the multi-objective targets?
- What is meant with "runs as normal"?
- What is optimized in between the query times?
- Section 2.3 mentions "assigning a fitness value" and "storing weight vectors". This suggest, the preferences are used to learn some feedback function that can be used to evaluate candidates. How does this differ from learning a fitness function?
- Unclear how "rounds" are related to the budget, timesteps and queries.

Furthermore, the experiment section also needs some additional polish:
- Results are not reproducible, as several hyperparameters are not mentioned, especially concerning the EA algorithms.
- Fig.3: How is it possible that RUCB-AL is outperforming RUCB after a single feedback round?
- "Our proposed method achieves the minimum mean in 8 instances" - This only seems true if one is able to optimize the EA algorithm used in conjunction with RUCB-AL. Of course, in practice this is not really possible. Under a fair comparison (preselected EA algorithm), IEMO/D seems to be the best?
- Table 1 hardly readable because of the number format. Maybe use relative values (e.g. relative difference to the optimum).
- Sec 3.4 not meaningful without the appendix, as one can not evaluate if the results are good or bad.
- PSP trials do not mentioned used budget, rounds, etc.

Additionally, the work is not sufficiently considering related work. E.g. "Preference-based Multiobjective Evolutionary Algorithm for Power Network Reconfiguration" or "A Preference Based Interactive Evolutionary Algorithm for Multi-objective Optimization". It should also be mentioned, that a vast number of query selection strategies already exist (e.g. see Bengs 2021), that aim at reducing the number of queries to be performed. Why should this not be considered "active learning"?

In conclusion, besides the clarity issues, two of the claims are not sufficiently substantiated:
- The approach does not seem to be fitness free, as Sec. 2.3 mentions something resembling learned functions.
- The difference between the "AL" part of algorithm and other query selection strategies.

**Questions:**

See weaknesses section.

---

> ### Author Response · Authors · 2023-11-21
> **Response to Reviewer 5t57 (part 1)**
>
> We are thankful for your comments. Your feedback has been immensely helpful in identifying areas for improvement and refinement in our work. We are committed to addressing these points to enhance the quality and comprehensiveness of our research.
>
> Regarding the highlighted weaknesses:
>
> 1. We’ve refined the clarity of this paper, the questions are answered as follows:
>     - The definition of $\hat P, P,\mathbf{w}$ are introduced in this paper, $P$ in basic definition, $\hat P, \mathbf{w}$ in **section 2.2.2**. The calculation of $\mathbf{w}$ is in **line 21** of **algorithm 1** which means we will increment the corresponding weight vector according to the comparison result. $w_{ij}$ denotes the total time arm i beat arm j. $P$ is the theoretical winning probability of incumbent solutions, the calculation of which is shown in **eq3**. $\hat P$ is the predicted value on $P$ defined by **eq4**. $\hat P$ rely on winning time matrix $\mathbf{w}$.
>     - We changed the reused symbols $L_t$ to $V_t$ (**section 2.4**) to avoid misdirection. The $V_t$ is a probability distribution of the whole population according to the recommended solutions while $\hat P$ denotes the winning probability of incumbent solutions (i.e., K arms).
>     - The definition of $p_{min}$ is introduced below **eq5** and in **line 5** in **Algorithm 1**.
>     - Firstly, the fitness function is not a function in $\mathbf{F}(x)$. It helps conduct non-dominated sort in MOEAs, especially dominance-based MOEAs. The fitness function can filter solutions generated by MOEAs and reserve non-dominated solutions. Decomposition-based and indicator-based MOEAs don’t need fitness function, they find solutions in every weight vector direction. So those weight vectors are not utilized to approximate fitness function. It is noteworthy we construct a virtual fitness function. State-of-the-art **preference-based evolutionary multi-objective (PBEMO)** algorithms are all model-based. They need to approximate the whole utility function which may not exist or be hard to express. Our virtual fitness function is only accurate concerning the optimal solution as **eq5** indicates.
>     - In our experiment session, preference information is produced by a **virtual user**, which in specific we will set a reference point as user preference. When comparing two incumbent solutions, we will compare their distance to the reference point.
>     - The virtual fitness function is actually $V_t$ (**section 2.4**).
> 2. About the combination of EAs and bandit algorithm:
>     - MOEAs will work with the bandit until they reach the maximum evaluation generation. The **bandit algorithm** performs as an **AI assistant** to guide the MOEAs to converge to user-preferred regions. Without the guidance of the bandit, MOEAs will converge to a PF rather than a certain solution.
>     - “runs as normal” means the MOEAs run as defined in their origin paper. We have refined this sentence.
>     - Between two queries, MOEAs will process the last recommendation. And the whole population will converge to the last recommendation iteratively.
>     - If I’m getting it right, the reviewer may misunderstand that our proposed method also needs to simulate a fitness function representing user preference.
>         - Traditional **PBEMO** algorithms learn user preference by **approximating a utility function** with the help of mathematical tools like GP, RankNet, etc. The utility function in reality may not exist or be hard to express. However, our proposed method learns the **optimal solution** with the help of a bandit. The virtual fitness function is only an **assumption** about preference distribution which is only accurate at global optima.
>         - “assigning a fitness value” means the fitness is assigned to solutions according to the virtual fitness function.
>         - We mentioned “storing weight vectors” in **section 2.4**. Those weight vectors are not used to approximate a fitness function. They are the corresponding weight vector of recommended solutions concerning **search direction**.
>     - In this paper, round shares the same meaning as timestep. In each round, the bandit algorithm will select two incumbent solutions to make the next query. Budget is the number of times the dueling bandit is allowed to present the query to the Oracle. The relationship between the round and budget is: $round > K\ge budget$.

---

> ### Author Response · Authors · 2023-11-21
> **Response to Reviewer 5t57 (part 2)**
>
> 3. About experiment:
>     - The results are reproducible and hyperparameter settings are introduced in **Appendix A.5.1**.
>     - As to why RUCB-AL outperforms RUCB after one round, this phenomenon is caused by the definition of RUCB-AL. In our proposed method, every element in the winning probability $p_{i,j,i\ne j}$ start from 0.5 while RUCB from 2 (as they define ${x\over 0}=1$). This setting makes RUCB run from a larger loss value.
>     - Although from the perspective of $Loss$, IEMO/D seems to have the most time of beating other algorithms. However, empirically we find only calculating $Loss$ is not enough. Because IEMO/D has the poorest convergence among these peer algorithms. The final results of IEMO/D are scattered around the whole PF. So apart from using $Loss$ as a performance metric, we intend to take the total $Loss$ of the whole population into consideration.
>     - We’ve rearranged **Table 1** and made it readable.
>     - **Section 3.4** gives an example of how our proposed method performs on PSP problems. As we can see from **Figure 6 (a)**, the red part denotes the native protein while the blue part denotes the predicted one. If the two overlap, this means our algorithm runs well on this protein structure. In this picture, we can see that our predicted result overlaps with the native one for the most part except at the tail. **Figure 6 (b)** shows our algorithm results from the perspective of energy. Our output population converges around the native protein energy. **Figure 6** suggests our proposed method can be implemented on PSP problems. More results are shown in **Appendix A.6**.
>     - The most important parameter of PSP problems is the native energy of different protein structures. The native energy of the 4 protein are listed in **Table 2**. The other parameter settings are aligned with Zhang et al. (2023). We added this statement to the paper.
>
>     Zhang, Zhiming, et al. "Pareto Dominance Archive and Coordinated Selection Strategy-Based Many-Objective Optimizer for Protein Structure Prediction." *IEEE/ACM Transactions on Computational Biology and Bioinformatics* (2023).
>
> 4. As to the three recommended reference papers, we’ve thoroughly read them.
>     - *“Preference-based Multiobjective Evolutionary Algorithm for Power Network Reconfiguration”* utilized a preference-based evolutionary multi-objective (PBEMO) algorithm, namely PD-NSGA-II, to solve power network reconfiguration problem. However, their proposed method is only an easily modified version of NSGA-II concerning the dominance relationship. It didn’t learn preference by approximating a utility function or through pairwise preference from human feedback. So we didn’t take this paper into account.
>     - *“A Preference Based Interactive Evolutionary Algorithm for Multi-objective Optimization: PIE”* introduced an interactive PBEMO algorithm implementing NAUTILUS. However, it is not based on pairwise preference learning which is different from our proposed method and peer algorithms in this paper. So we didn’t take this paper’s method as a peer algorithm.
>     - *“Preference-based Online Learning with Dueling Bandits: A Survey”* listed state-of-the-art dueling bandit algorithms. As far as we investigated, none of these dueling bandit algorithms combine active learning in their process. They referred to active ranking, but this method is not a dueling bandit algorithm.
> 5. As to two of the claims:
>     - Our proposed method is fitness-free because we don’t learn a fitness function through the process of preference learning. Compare to those fitness-based PBEMO algorithms (e.g., I-EMO/D-PLVF, I-EMO/D-PPL) that **approximate a fitness (utility) function** in preference learning, our method only cares about the **global optima**. Then construct a **virtual utility function** that is only accurate at global optima. This process avoids approximating a fitness function that may not exist or is hard to express in reality.
>     - Other query strategies balance between exploration and **exploitation from the perspective** of different kinds of **regret**. However, our proposed method balances between **random search and greedy search** from the perspective of **loss (uncertainty)**. With the help of active learning, our proposed method can prompt the dueling bandit’s ability to progressively find the global optima within a limited budget. This is very important in real problems, because users may get impatient and give inconsistent preference after querying too much.
>
> We deeply appreciate your meticulous review and insightful suggestions. Should there be additional recommendations or insights you might offer, we would eagerly welcome them. Your time and dedication to evaluating our work are sincerely valued.

---

> ### Comment · Reviewer_5t57 · 2023-11-30
>
> Thanks for the extensive reply and clarifications. It seems you improved the paper greatly. However, the changes to the paper had been to substantial to review again that short before the end of the rebuttal phase. Additionally, it seems the new passages need another round of proof reading, as they contain quite a few wording/spelling mistakes. Furthermore, I'm still not convinced that your approach is utility free, because of Eq.9. In conclusion, i'm considering to improve my score slightly, but this will not change the overall picture, as it seems.

---

### Official Review · Reviewer_ch6a · 2023-10-27

**Soundness:** 4 excellent
**Presentation:** 3 good
**Contribution:** 2 fair
**Rating:** 3
**Confidence:** 4

**Summary:**

This paper proposes a new framework that firstly learns the objective of human users by dueling bandits, and then the learned objective can be optimized by some zero-th order optimization methods. Specifically, the authors first propose an active dueling bandits algorithm, in which the feedback obtained from the comparisons is used to update the preference probability estimates for each solution, which are then used to construct the user-specific objective function. Based on the objective funciton, the second step is to use the learned objective function to guide the optimization process in a multi-objective evolutionary algorithm. The EA uses the learned objective function to evaluate the fitness of candidate solutions and to guide the search towards the region of interest along the Pareto frontier.

**Strengths:**

- The problem this paper concerns is clearly defined, and all formalizations are clean and easy to follow

- The paper is well-organized with clear writing.

- Real-world application on protein structure prediction is conducted, which may be closely related to AI for science.

**Weaknesses:**

- My major concern is the novelty of the proposed framwork, as learning objective functions according to human feedback is not new[1][2][3].

Ref.

[1] Roijers, Diederik M., Luisa M. Zintgraf, and Ann Nowé. "Interactive thompson sampling for multi-objective multi-armed bandits." Algorithmic Decision Theory: 5th International Conference, ADT 2017
[2] Ding, Yao-Xiang, and Zhi-Hua Zhou. "Preference based adaptation for learning objectives." Advances in Neural Information Processing Systems 31 (2018).
[3] Tucker, Maegan, et al. "Preference-based learning for exoskeleton gait optimization." 2020 IEEE international conference on robotics and automation (ICRA). IEEE, 2020.

- The proposed active dueling bandits algorithm is confusing. In traditional active learning, there should be an uncertainty measure, according to which the learner decides whether to query; in active dueling bandits proposed in this paper, if I'm getting it right, whether to query if sorely focusing on if the pair is compared before, which is a noisy feedback that is not trustworthy. I'm not saying it's wrong, considering the Copeland Winner's assumption that the pairwise winning probability is not independent but with some internal structure that can be used, but it should be discussed why such active query mechanism is reasonable under the Copeland setting. Besides, the regret if RUCB-AL is linear of T, which means it does not converge to the Copeland winner.

**Questions:**

See weaknesses above.

---

> ### Author Response · Authors · 2023-11-21
> **Response to Reviewer ch6a (part 1)**
>
> We are thankful for your comments. Your feedback has been constructive in identifying areas for improvement and refinement in our work. We are committed to addressing these points to enhance the quality and comprehensiveness of our research.
>
> Regarding the highlighted weaknesses:
>
> 1. The reviewer may misunderstand our research field. If I’m getting it right, the reviewer thinks our proposed method is to conduct preference learning from human feedback and also learn an objective function representing user preference. However, we are implementing our proposed method in a **preference-based evolutionary multi-objective optimization (PBEMO)** scenario.
>     - As far as we know, we are the first to adopt fitness-free preference learning in MOEAs. State-of-the-art algorithms learn user preference by approximating a utility function, which in reality may not exist or is hard to express. This kind of method is called fitness-based or model-based because it depends on prior knowledge of the distribution of solutions (e.g., Bradley Terry model). However, our proposed method utilizes the property of dueling bandit to conduct preference learning in MOEAs. Given incumbent solutions, the dueling bandit will not try to approximate a utility function. Conversely, it only focuses on the optimal solution. To utilize the recommended solution $\hat{\mathbf{z}}^*$, we construct a virtual utility function. The virtual utility function $V_t$ (**section 2.4**) is only accurate regarding optimal solutions.
>     - As to the three recommended reference papers, we’ve thoroughly read them. The research scenarios are different. **Roijers et al. (2017)** studied preference learning in an online interactive multi-objective reinforcement learning (MORL) scenario. They mapped a policy to an arm and with the help of a dueling bandit the policy can converge to an optimal solution aggressively. **Ding & Zhou (2018)** targeted learning a linear combination loss function that outperforms other loss functions. The different loss ingredients are so-called multi-objectives. Their learned loss function will participate in the optimization process. The preference learning indirectly helps optimization. **Tucker et al. (2020)** proposed a Bayesian dueling bandit method for single-objective preference learning. These three research all converge to **a single solution or an aggregation** at last. However, in our research field, PBEMO, preference learning directly helps the optimization algorithm converge to **a set of favorite solutions**. Our final results are a population that helps our algorithm avoid trapping in local optima.
> 2. About the proposed active dueling bandit method:
>     - For our proposed method, we have uncertainty measures to decide whether to query. As can be seen in **eq5**, we decide the next query based on a balance between random and greedy search. Also, our proposed dueling bandit is not just based on whether it has been asked before. The two arms are selected according to the definition of $p_{a_c}$ in **eq5**. Then if the two have been queried before, they will not be presented to the user again.
>     - We understand your concern that user feedback may be noisy and not trustworthy. In our experiment, the user feedback follows the **Bernoulli distribution**. It doesn’t matter in this consultation session user gives the wrong answer. Our preference will be adjusted in several consultations.
>     - About the regret bond, we are confident that our regret bond can be polished up since the experiment results suggest our proposed method works well with multi-objective optimization problems. We will polish it up and reference CCB and SCB structure (Zoghi et al., 2015).
>
> We deeply appreciate your meticulous review and insightful suggestions. Should there be additional recommendations or insights you might offer, we would eagerly welcome them. Your time and dedication to evaluating our work are sincerely valued.

---

### Official Review · Reviewer_NX1i · 2023-10-29

**Soundness:** 3 good
**Presentation:** 2 fair
**Contribution:** 2 fair
**Rating:** 3
**Confidence:** 3

**Summary:**

The authors study the problem of optimizing multi-objectives.  Instead of calculating the fitness function, authors leverage a human preference-type algorithm, motivated by dueling bandits, to solve their problem. The authors further apply their algorithm to practical applications.

**Strengths:**

1. The authors propose a novel framework to return the optimized objective by using a dueling bandit framework. This motivation is great since, in reality, human's preferences are easier to obtain than a real-valued reward.

2. The authors derived theoretical guarantee for the dueling bandit algorithm and applied it to protein structure prediction.

**Weaknesses:**

Main weakness:

1. The result of the theories seems weak. The authors derive this algorithm in an online version, but the regret order is O(T), which is too large and not efficient.

2. Weak connections between the dueling bandit algorithm and the main goal of the paper. It seems only this regret is analyzed instead of how well it optimizes the objectives, which are not discussed in a theoretical manner.

3. Lack of comparison with benchmarks from both dueling bandit and the main optimization objective in the paper

**Questions:**

1. The authors provide a regret for the proposed algorithm with order $O(T)$. This result seems too weak. Could the author further improve this bound? Also, could the authors compare this with some existing benchmarks? Since there are many algorithms that are available in dueling bandit area that can achieve $O(\sqrt{T})$ upper bound

2. When implementing the dueling bandit, it seems the greed algorithm was implemented. What if we consider an algorithm with an upper confidence interval? Would this give a better regret bound?

3. There is no clear and explicit connection between the dueling bandit algorithm and the key objective of this paper, could the authors provide some theoretical guarantees on this?

---

> ### Author Response · Authors · 2023-11-21
> **Response to Reviewer NX1i (part 1)**
>
> We are thankful for your comments. Your feedback has been constructive in identifying areas for improvement and refinement in our work. We are committed to addressing these points to enhance the quality and comprehensiveness of our research.
>
> Regarding the highlighted weaknesses and questions:
> 1. We are confident that our regret bond can be polished up since the experiment results suggest our proposed method works well with multi-objective optimization problems. We are committed to considering CCB and SCB structure (Zoghi et al., 2015) and further refining the regret bond associated with our proposed method.
> 2. We added an elaboration of the optimization module (**section 2.2**) before the preference elicitation module to connect our proposed method with optimization. Since not all feasible solutions are visible to users, MOEAs perform as optimizers to generate incumbent solutions.
>     - In this paper, we haven’t considered MOEAs to utilize the regret. In our future work, we are going to utilize the regret in MOPs (e.g., take the recommendation point $\hat {\mathbf{z}}^*$ as a distribution rather than a converging target).
>     - As to why not use the UCB function. Traditional UCB is not active enough, empirically given $K=10$ we will need more than 300 rounds to receive an acceptable accuracy. Our proposed method is inspired by UCB. Traditional RUCB algorithms only balance the mean and uncertainty of winning probability. Our designed active RUCB-AL balances exploration and exploitation from another perspective. We've elaborated on the components of **eq 5**, emphasizing its role in achieving a balance between random and greedy search strategies.
> 3. Since the key of this paper is **preference-based multi-objective optimization (PBEMO)**, it is natural we focus more on multi-objective optimization. The single-objective experiment is only designed to prove its convergence. We have sufficient multi-objective experiments. Our proposed method compared with **6 peer algorithms** on **13 benchmark** problems with versatile PF shapes and dimensions ($m=\{2,3,5,8,10\}$). Additionally, we conducted our method on a complicated real-world problem, PSP.
>
> We deeply appreciate your meticulous review and insightful suggestions. Should there be additional recommendations or insights you might offer, we would eagerly welcome them. Your time and dedication to evaluating our work are sincerely valued.

---

### Official Review · Reviewer_Xsp2 · 2023-11-02

**Soundness:** 1 poor
**Presentation:** 1 poor
**Contribution:** 1 poor
**Rating:** 3
**Confidence:** 3

**Summary:**

The goal of this paper is to guide evolutionary algorithms via human preference feedback, so that there is no need to design the fitness function. The authors propose to achieve this by adopting a modified version of the dueling bandit algorithm, RUCB (Zoghi et al. 2014).

**Strengths:**

The idea of combining dueling bandits and evolutionary algorithms is novel and well-motivated, as it naturally allows using human preference feedback to guide evolution.

**Weaknesses:**

1. The presentation of this paper needs significant improvement. Currently, the problem setting, the objective, and the description of the proposed "human-dominated preference-based EMO" are not explained clearly, with several important details being vague/missing.

For example, there lacks a clear and rigorous formulation of the problem illustrated in Figure 1. The author only provided formulation for the "consultation module" in Section 2.2.1, which explains when given a set of individuals, the goal of this module is to find the Copeland winner based on human preference feedback. However, it is not clear why we need the evolutionary algorithm to optimize Eq 1.
I inferred from the context that, the goal of the whole pipeline is to find the optimal solution of Eq 1. But the learner cannot simply choose the solution in set $\Omega$ to maximize function $F$, i.e., the learner cannot observe the whole set $\Omega$. Instead, it has to rely on a evolutionary algorithm to create new solutions. Please correct me if I misunderstood anything.

Also, it is also quite confusing to me that the multi-objective in Eq 1 plays no role in the formulation of dueling bandit, i.e., how to compare two arms under multi-objective setting.

2. RUCB algorithm relies on the assumption that there exists a Condorcet winner. Relaxation of this assumption, i.e., find Copeland winner instead, requires different algorithm design, like the CCB and SCB algorithm in Zoghi et al. (2015). The authors dropped the Condorcet winner assumption in Assumption 1. I would appreciate it if the authors could elaborate on how this can be achieved with the modified RUCB algorithm in Algorithm 1.

Zoghi, M., Karnin, Z.S., Whiteson, S. and De Rijke, M., 2015. Copeland dueling bandits. Advances in neural information processing systems, 28.

3. Related to the previous comment, it is not clear to me why the bonus term of standard RUCB algorithm is modified to Eq 5. In addition, if I am not missing anything, the squared loss of $\hat{p}\_{i,j}$ in Eq 5 cannot be computed by the algorithm, since the value of $p_{i,j}$ is unknown?

**Questions:**

The definition of Copeland winner in Eq 2 does not seem to be correct. Moreover, as Copeland winner is guaranteed to exist, it would be better to call this as "Definition" instead of "Assumption".

---

> ### Author Response · Authors · 2023-11-21
> **Response to Reviewer Xsp2**
>
> Thank you sincerely for the invaluable insights and thoughtful comments on our paper. Your feedback has been constructive in identifying areas for improvement and refinement in our work. We are committed to addressing these points to enhance the quality and comprehensiveness of our research.
> Regarding the highlighted weaknesses:
> 1. About clarity:
>     - We have included a dedicated **section 2.2** elucidating the optimization module's significance, particularly emphasizing the necessity of employing multi-objective optimization algorithms (MOEAs). Since not all feasible solutions are visible to users. MOEAs play the role of generating solutions that are actively sampled and fed into bandit for preference learning. We've clarified that our research problem operates within multi-objective scenarios, especially **preference-based evolutionary multi-objective optimization (PBEMO)**.
>    - Traditional PBEMO algorithms learn user preference by learning fitness. However, such a fitness function may not exist or be hard to express. Also, traditional methods are model-based for they depend on different mathematical models (i.e., Bradley Terry model) which requires a lot of expert knowledge about the environment. Our proposed fitness-free PBEMO architecture does not need to simulate a fitness function and the virtual fitness function is only accurate about optimal solutions. Also, our method is model-free. We are not based on any mathematical model assumptions.
>     - Additionally, we've explained our utilization of the widely adopted **sigmoid function** to map multi-objective solutions to arms, establishing a link between recommended solutions and a virtual utility function ($V_t$) for preference probability computation $p_{ij}=sigmoid(V_t(\mathbf{z}_i)-V_t(\mathbf{z}_j))={1\over 1+e^{-(V_t(\mathbf{z}_i)-V_t(\mathbf{z}_j))}}$.
> 2. We have corrected the definition of the Copeland winner (**eq 2**). In order to actively query users, we designed our bandit by choosing arms from the perspective of uncertainty (i.e., MSE loss). We believe by improving the certainty of the winning probability matrix, the bandit algorithm will output a more precise Copeland winner within a limited budget. Empirically, this method proves to be efficient. We are also committed to considering CCB and SCB structure (Zoghi et al., 2015) and further refining the regret bond associated with our proposed method.
> 3. Traditional RUCB only balances the mean and uncertainty of winning probability. Our designed active RUCB-AL balances exploration and exploitation from another perspective. The first term in **eq5** refers to a decaying random search process, while the second term refers to a greedy search based on the loss function. We designed the **eq5** to balance between random search and greedy search. As to how to achieve $p_{i,j}$ which is unknown to the user. It is approximated with the following steps:
>     - In the first round, $p_{i,j}=0.5,\ \forall\ i,j\in K$,
>     - After the second round, we will receive a current optimal solution $\hat{\mathbf{z}}^*_t$,
>     - According to the recommendation $\hat{\mathbf{z}}^*_t$, we can calculate a virtual utility function $V_t$ (**section 2.4**). Unlike existing PBEMOs that require mathematical tools like GP to approximate the whole utility function, our virtual utility function is not accurate except at the optimal solution.
>     - calculate $p_{i,j}=\sigma(V_t(\mathbf{z}_i)-V_t(\mathbf{z}_j))$
>
> We deeply appreciate your meticulous review and insightful suggestions. Should there be additional recommendations or insights you might offer, we would eagerly welcome them. Your time and dedication to evaluating our work are sincerely valued.

---

### Meta-Review · Area_Chair_CsiS · 2023-12-05

**Metareview:**

This paper proposes a novel approach to solving multi-objective problems where the human feedback guides evolutionary algorithms. The idea is natural and has potential. The shortcomings of the paper are:

* **Clarity:** Presentation needs major improvements. Notation is poorly defined and ambiguous.

* **Analysis:** Linear regret. This generally indicates that the analysis is extremely loose.

* **Prior work:** As mentioned by Reviewer ch6a, there are many prior works on interactive methods for multi-objective optimization. Better discussion of prior works and including them in experiments is necessary.

The authors tried to close these gaps in the rebuttal but this was too much to overcome.

**Justification For Why Not Higher Score:**

This paper was clearly on the reject side.

**Justification For Why Not Lower Score:**

N/A

---

### Decision · Program_Chairs · 2024-01-16

Reject